# Specific residues in the cytoplasmic domain modulate photocurrent kinetics of channelrhodopsin from *Klebsormidium nitens*

Rintaro Tashiro [1], Kumari Sushmita [2], Shoko Hososhima [1,3], Sunita Sharma [2], Suneel Kateriya [2], Hideki Kandori [1,3] & Satoshi P. Tsunoda [1,3 ✉]

Channelrhodopsins (ChRs) are light-gated ion channels extensively applied as optogenetics tools for manipulating neuronal activity. All currently known ChRs comprise a large cytoplasmic domain, whose function is elusive. Here, we report the cation channel properties of KnChR, one of the photoreceptors from a filamentous terrestrial alga *Klebsormidium nitens*, and demonstrate that the cytoplasmic domain of KnChR modulates the ion channel properties. KnChR is constituted of a 7-transmembrane domain forming a channel pore, followed by a C-terminus moiety encoding a peptidoglycan binding domain (FimV). Notably, the channel closure rate was affected by the C-terminus moiety. Truncation of the moiety to various lengths prolonged the channel open lifetime by more than 10-fold. Two Arginine residues (R287 and R291) are crucial for altering the photocurrent kinetics. We propose that electrostatic interaction between the rhodopsin domain and the C-terminus domain accelerates the channel kinetics. Additionally, maximal sensitivity was exhibited at 430 and 460 nm, the former making KnChR one of the most blue-shifted ChRs characterized thus far, serving as a novel prototype for studying the molecular mechanism of color tuning of the ChRs. Furthermore, KnChR would expand the optogenetics tool kit, especially for dual light applications when short-wavelength excitation is required.

[1] Department of Life Science and Applied Chemistry, Nagoya Institute of Technology, Nagoya, Japan. [2] Laboratory of Optobiology, School of Biotechnology, Jawaharlal Nehru University, New Delhi, India. [3] OptoBioTechnology Research Center, Nagoya Institute of Technology, Showa-Ku, Nagoya, Japan. ✉email: tsunoda.satoshi@nitech.ac.jp

C channelrhodopsins (ChRs) are directly light-gated ion channels naturally found in the eyespot of green algae[1–3]. Channelrhodopsin-1 and -2 from *Chlamydomonas reinhardtii* (CrChR1 and CrChR2) were the first ChRs to be discovered, characterized, and the latter was extensively utilized for optogenetic applications. These proteins conduct cations such as $H^+$, $Na^+$, $K^+$, and $Ca^{2+}$. High-resolution X-ray structures of ChRs revealed details of the molecular architecture and provided insight into the photoactivation and ion conduction pathway[4,5]. ChRs have been applied to generate action potentials in the light-insensitive cells and tissues with unprecedented spatio-temporal precision, which initiated a new research field, optogenetics[6,7]. ChR variants have been engineered to improve the functionality, and homologous ChRs were also reported accordingly[8]. Anion-conducting ChRs (ACRs) have been created artificially or naturally discovered[9–11]. These proteins are applied in neuroscience to optically silence action potential.

Recently, the genome of terrestrial green alga (*Klebsormidium flaccidum*) was sequenced, and analysis of the genome delineated the molecular basis of the adaptation of land plants[12]. *K. flaccidum* was renamed as *Klebsormidium nitens* (http://www.plantmorphogenesis.bio.titech.ac.jp/~algae_genome_project/klebsormidium/). *K. nitens* adapts to diverse environmental conditions such as desiccation, light, and temperature gradients[13]. However, the sensory photoreceptor landscape of *K. nitens* has not yet been established. For the first time, we identified a gene encoding a putative channelrhodopsin from *K. nitens* (KnChR). A unique feature of KnChR is that it includes the FimV domain at the C-terminus of rhodopsin. FimV is a peptidoglycan-binding protein that is responsible for twitching motility in *Pseudomonas aeruginosa* and other microbes[14]. All so far known ChRs comprise a few hundreds of amino acid residues in their cytoplasmic domains, for example, full-length CrChR1 and CrChR2 contain 712 and 737 amino acids respectively[2,3] of which the 7-transmembrane (TM) rhodopsin domain is made up of only about 300 amino acids. Thus, 400–450 amino acids are located mainly on the cytoplasmic side. However, no significant homology was found among the long cytoplasmic extension of the characterized ChRs from different organisms. Since the discovery of ChRs, the role of cytoplasmic domains remains elusive. The initial reports of CrChR1 and CrChR2 described that truncation of the cytoplasmic C-terminal domain did not alter the ion channel function[2,3]. Thus, researchers investigated their molecular properties by using truncated proteins carrying only the 7-TM domain. However, an extensive study of the phosphoproteome of the eyespot fraction of *C. reinhardtii* showed three phosphorylation sites for CrChR1 and one for CrChrR2[15]. Thus, a kinase-dependent regulation of function was also proposed. Recently, it has been shown that phosphorylation of CrChR1 regulates photomotility and calcium signaling of green alga[16].

In this report, we performed extensive electrophysiological experiments using KnChR to reveal its ion channel function. Its photocurrent properties such as channel kinetics, absorption maxima, light sensitivity, and ion selectivity were compared to those of well-characterized CrChR2. For the first time, we show an unexpected modulation of the photocurrents of KnChR by its extended C-terminus region. Furthermore, KnChR has potential for expanding the optogenetics tool kit, especially for dual light applications when short-wavelength excitation is required. Moreover, functional characterization of ChR from an evolutionary terrestrial alga indicates the existence of ChR-mediated physiological responses beyond aquatic habitats (marine and freshwater systems).

## Results

Mining of the genomic database of *K. nitens* revealed existence of several rhodopsin-encoding genes (http://www.plantmorphogenesis. bio.titech.ac.jp/cgibin/blast/blast_www_klebsormidium.cgi). Among them, we found one gene that is highly homologous to chlorophyte cation ChR such as CrChR2. We named it KnChR in this study. KnChR possesses 831 amino acids and harbors a membrane-embedded 7-TM rhodopsin domain (amino acids 1–272) followed by a long C-terminus region (273–831 amino acids) (Supplementary Fig. 1). Sequence comparison of the rhodopsin domain revealed 34.5% identity and 69.3% homology with CrChR2. Several important amino acid residues were aligned when compared to four ChRs and bacteriorhodopsin (BR) (Fig. 1a). Three (Glu82, Glu90, Glu97) out of five glutamate residues at positions 82, 83, 90, 97, and 101 in CrChR2, important for channel activity are conserved in KnChR, whereas amino acid corresponding to Glu83 and Glu101 in CrChR2 are replaced by Val and Pro in KnChR respectively. Residue Asp156 and Cys128 in CrChR2 form a hydrogen bridge (D-C pair or D-C gate) which mediates the channel open lifetime[17,18]. This pair is conserved in KnChR. Thus, we predicted that KnChR might function as a cation channelrhodopsin when functionally expressed in a suitable system. Notably, KnChR has a long C-term domain that encodes a putative peptidoglycan binding domain (FimV) between 410–690 amino acids (Supplementary Fig. 1). In general, the peptidoglycan layer is located outside of the cytoplasmic membrane in bacterial cell walls. In fact, the homologous region of KnChR is predicted to be located on the extracellular side, whereas the C-terminus region is normally located on the cytoplasmic side in the case of microbial rhodopsin (Supplementary Fig. 2). Thus, we considered why the FimV domain is connected to the C-terminus of the 7-TM domain. We anticipated that the membrane topology of KnChR is inverted, as in the case of heliorhodopsin (HeR), which is a recently identified subfamily in microbial rhodopsins[19]. We synthesized a codon-adapted full-length gene of KnChR (831 amino acids) for functional expression in a mammalian system and fused it with the p3.0-eYFP vector with eYFP at the C-terminus of KnChR. The vector contains the signal peptide for export to ER and membrane trafficking to improve membrane localization. After transfection of the fusion expression construct, we observed weak eYFP fluorescence on the cellular membrane, indicating poor expression in ND7/23 cells. A voltage-clamp measurement showed no photocurrent. We anticipated that low expression was due to a long C-terminus domain (~540 amino acids) after the 7-TM domain. Thus, we systematically truncated the C-terminus to various lengths. We created five variants with different lengths, carrying 697, 397, 310, 290, and 272 amino acids. Note that only the variant with 697 amino acids included the FimV domain. In addition, we fused a c-Myc epitope tag at the C-terminus of the 697 amino acids variant to detect immune fluorescence. Figure 1b-d shows fluorescent images of the transfected cells. eYFP fluorescence indicates the successful expression of KnChR. Although no alexa595 signal was observed in the absence of detergent (Fig. 1b), treatment with detergent to permeabilize the membrane generated a signal (Fig. 1c). No alexa595 signal was observed when KnChR without c-Myc-tag was expressed even after the detergent treatment (Fig. 1d). These results indicate that the C-terminus domain is on the cytoplasmic side. Thus, KnChR exhibits a typical membrane topology of a channelrhodopsin.

**The C-terminus of KnChR modulates photocurrents of the channelrhodopsin domain**. We then performed patch-clamp recordings of each variant in detail. All the constructs having various amino acid lengths are depicted in Fig. 2a. As shown in Fig. 2 b–f, photocurrents were observed in all the five KnChR variants upon 480 nm light illumination for 1.0 s. Photocurrent amplitudes were comparable to that of CrChR2 (Fig. 2g). The current direction reversed as clamped voltage shifted from −90 to +50 mV in 20 mV steps, which is a typical ion channel property. Illumination induced a peak current (Ip) which slowly decayed

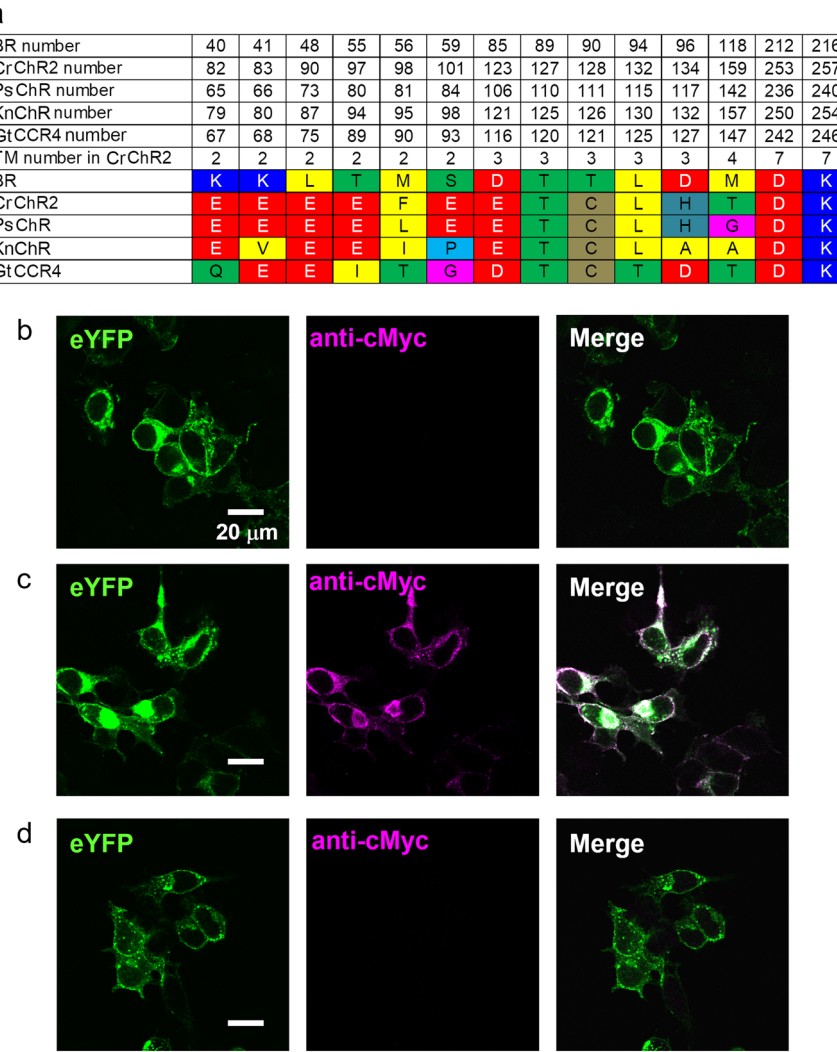

| BR number | 40 | 41 | 48 | 55 | 56 | 59 | 85 | 89 | 90 | 94 | 96 | 118 | 212 | 216 |
|---|---|---|---|---|---|---|---|---|---|---|---|---|---|---|
| CrChR2 number | 82 | 83 | 90 | 97 | 98 | 101 | 123 | 127 | 128 | 132 | 134 | 159 | 253 | 257 |
| PsChR number | 65 | 66 | 73 | 80 | 81 | 84 | 106 | 110 | 111 | 115 | 117 | 142 | 236 | 240 |
| KnChR number | 79 | 80 | 87 | 94 | 95 | 98 | 121 | 125 | 126 | 130 | 132 | 157 | 250 | 254 |
| GtCCR4 number | 67 | 68 | 75 | 89 | 90 | 93 | 116 | 120 | 121 | 125 | 127 | 147 | 242 | 246 |
| TM number in CrChR2 | 2 | 2 | 2 | 2 | 2 | 2 | 3 | 3 | 3 | 3 | 3 | 4 | 7 | 7 |
| BR | K | K | L | T | M | S | D | T | T | L | D | M | D | K |
| CrChR2 | E | E | E | E | F | E | E | T | C | L | H | T | D | K |
| PsChR | E | E | E | E | L | E | E | T | C | L | H | G | D | K |
| KnChR | E | V | E | E | I | P | E | T | C | L | A | A | D | K |
| GtCCR4 | Q | E | E | I | T | G | D | T | C | T | D | T | D | K |

**Fig. 1 Sequence comparison and determination of KnChR transmembrane topology and orientation. a** Amino acid alignments of bacteriorhodopsin (BR), CrChR2 (Channelrhodopsin-2 from *Chlamydomonas reinhardtii*), PsChR (Channelrhodopsin from *Platymonas subcordiformis*), KnChR and GtCCR4 (Cation Channelrhodopsin-4 from *Guillardia theta*). The characteristic amino acids in BR and CrChR2 were selected. In addition, amino acid numbers of each protein and transmembrane helix (TM) number are indicated. See Supplementary Fig. 1 for an alignment of the whole protein sequences. **b** Immunostaining of KnChR. Expression of KnChR-3.0-eYFP-cMyc in cultured ND7-23 cells, with KnChR bearing the c-Myc epitope tag at the C-terminus. eYFP fluorescence (left, green), probed with a c-Myc antibody under non-permeabilized conditions for immunofluorescent staining with Alexa Fluor 594 (middle, magenta) and merge (right). Scale bar, 20 μm. **c** Expression of KnChR-3.0-eYFP-cMyc in cultured ND7-23 cells. eYFP fluorescence (left, green), probed with a c-Myc antibody under permeabilized conditions (0.5% Triton X-100) for immunofluorescent staining with Alexa Fluor 594 (middle, magenta) and merge (right). **d** Expression of KnChR-3.0-eYFP in cultured ND7-23 cells. eYFP fluorescence (left, green), probed with a c-Myc antibody under permeabilized conditions (0.5% Triton X-100) for immunofluorescent staining with Alexa Fluor 594 (middle, magenta) and merge (right).

into lower amplitudes during illumination. However, the current did not reach a plateau within 1 s, whereas the photocurrent of CrChR2 rapidly dropped to a steady-state level (Fig. 2g). When the light was shut off, this diminished the photocurrent into the baseline, indicating channel closure. Notably, the phases of current decay were significantly different among the five variants, two of which, those with 697 and 397 amino acids, rapidly decayed into the baseline whereas the other three (300, 290, and 272 amino acids) showed a relatively slow decay (Fig. 2b–f). This indicates that the channel kinetics is affected by the cytoplasmic domain of KnChR. To further investigate the above observation, we created in addition four variants comprising different numbers of amino acid residues: 321, 317, 310, and 280. Current decay kinetics (τ-off) at −70 mV from all nine variants were compared (Fig. 2h). The variants longer than 317 amino acids showed a fast off-kinetics (~10 ms). The variant with 310 amino acids showed

an off-kinetics of about 20 ms. Furthermore, the time constants of the shorter variants became even larger and the shortest variant with 272 amino acids reached 130 ms, more than 10 times slower than the original construct. This result clearly shows a correlation between the length of amino acids and the time constant of channel kinetics in KnChR, indicating that the C-terminal cytoplasmic domain affects channel properties. In particular, amino acids between 272 and 310 largely contribute to the modulation of light-gated ion channel activities in KnChR.

The photocurrent amplitudes also differed among the nine variants (Fig. 2i). The amplitude of the variants shorter than 300 amino acids reached about 3 nA at −70 mV, while the longer variants >310 amino acids showed smaller currents under the same condition in which only about 1 nA was observed from cells expressing the 397 and 697 amino acids variants. We reasoned that the current amplitude depends on the expression levels of

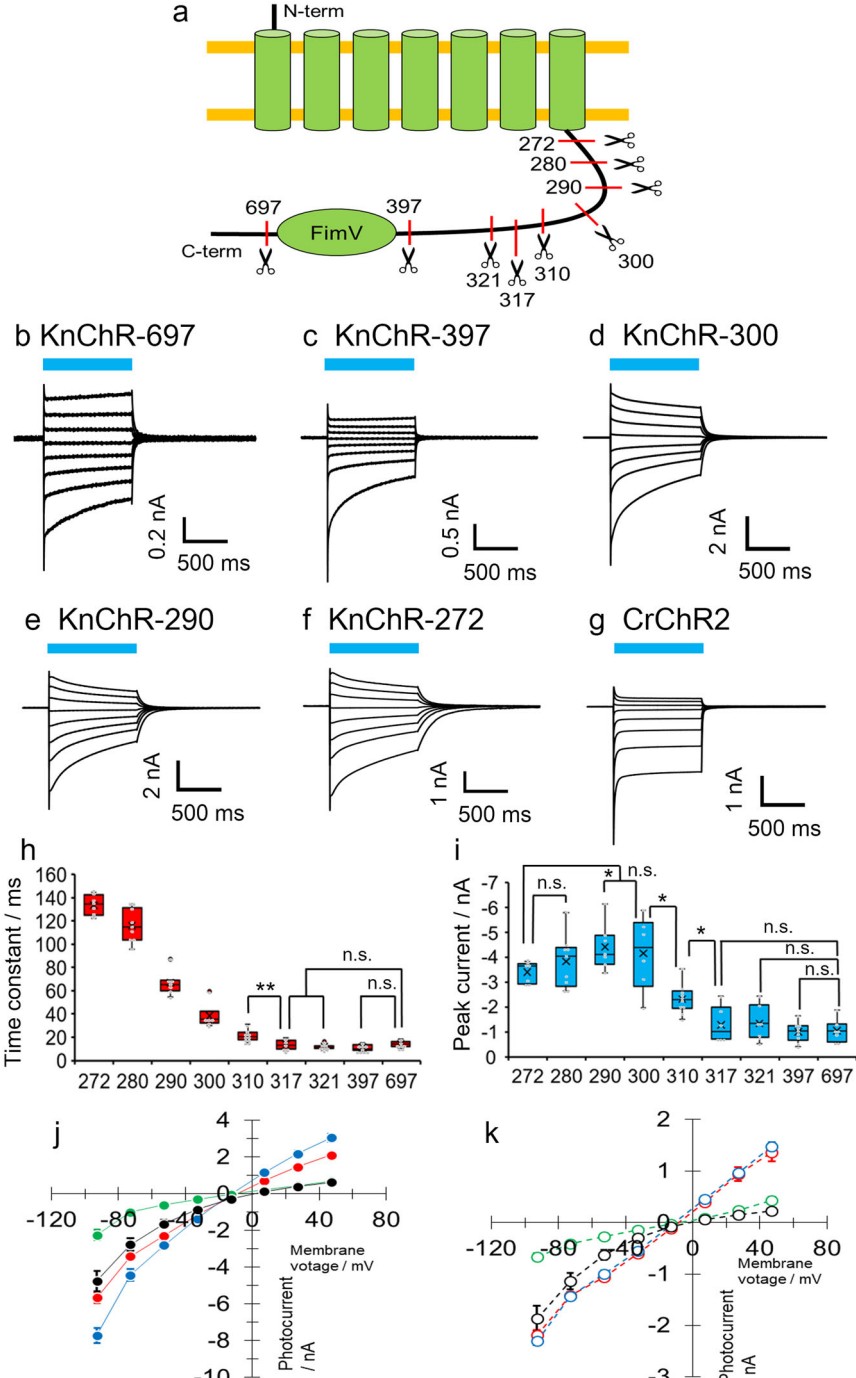

**Fig. 2 Electrophysiological measurements of various C-terminal length constructs of KnChR. a** Schematic drawing of truncated constructs. Amino acid positions truncated are indicated. **b–g** Photocurrent traces of various C-terminal length constructs and CrChR2. Blue bars indicate light application (480 nm, 12.3 mW/mm$^2$). Membrane voltage was clamped from at −90 to +50 mV by 20 mV steps. Standard solutions were used (See "Methods"). **h** and **i** Comparison of time constants and photocurrent amplitude from KnChR variants. The membrane voltage was clamped at −70 mV. N = 7 (272), 9 (280), 7 (290), 6 (300), 10 (310), 6 (317), 6 (321), 9 (397), and 7 (697). The box-and-whisker plot represents the median (center line), the mean (x), interquartile range (box limits) and 1.5 × interquartile range (whiskers). **j, k** I-V relationship of KnChR and CrChR2. Filled and empty symbols indicate peak and steady-state current, respectively. Red: KnChR-272 (N = 7), Blue: KnChR-290 (N = 7), Green: KnChR-397 (N = 9), Black: CrChR2 (N = 6). Data were presented as the mean ± SEM.

each variant. Longer constructs are generally known to show lower expression levels. Indeed, we observed only poor expression of the full-length construct as mentioned above. Therefore, the poorer expression seems to be the most likely reason for smaller currents recorded from the longer constructs as compared with the shorter ones.

The current-voltage relations (I-V plots) of KnChR and CrChR2 are depicted in Fig. 2j (peak component) and Fig. 2k (steady-state component). The current direction and amplitude depend on the applied voltage. The reversal potential of KnChR is about −10 mV while that of CrChR2 is 0 mV under the same condition. As is already known, ion conductance of CrChR2 is

inwardly rectified and thus the outward current was largely suppressed (Fig. 2j and k, black).

To compare rectification, I-V plots of CrChR2 and KnChR were created after normalizing the current amplitude (at +50 mV as 1.0) (Supplementary Fig. 3). When the absolute magnitude at +50 and −50 mV are compared, the photocurrent of KnChR at −50 mV is −1.085, indicating an almost linear relation between +50 and –50 mV. On the other hand, the photocurrent of CrChR2 at −50 mV is −2.6, showing inward rectification. These results indicate that two channelrhodopsins exhibit different rectification. The I-V plots of the five variants are summarized in Supplementary Fig. 4. The photocurrent properties of KnChR 397 amino acids (excluding FimV) and 697 amino acids (including FimV) are shown in Supplementary Fig. 5. The current shape, I-V relation, and current amplitudes of these two variants were almost identical. These results indicate that FimV did not alter channel function.

All nine variants that were tested were fused to eYFP at the C-terminal. Thus, one might argue whether the length-dependent effect in Fig. 2h, i might have originated from eYFP, i.e., if an interaction between rhodopsin and eYFP may have altered the channel kinetics for an unknown reason. To exclude this possibility, we created 272 amino acids and 397 amino acids KnChR variants without eYFP. The variants were transfected, and the photocurrents were recorded to compare the channel-off kinetics and amplitude (Supplementary Fig. 6). The time constants (τ-off) were almost identical between eYFP-tagged and untagged 272 amino acids and 397 amino acids variants (Supplementary Fig. 6a). This clearly indicates that a length-dependent effect was observed, regardless of the presence of eYFP at the C-terminus of KnChR. Photocurrent amplitudes of untagged variants were smaller than those of eYFP-tagged constructs (Supplementary Fig. 6b). This could be due to a lower expression level of the former.

**Positively charged residues in the C-terminus domain of KnChR regulated intramolecular interactions.** Based on the result in which the cytoplasmic domain altered the channel kinetics in KnChR (Fig. 2h), it could be hypothesized that the cytoplasmic domain interacts with the rhodopsin domain (7-TM domain). Since no significant change in τ-off was observed among variants longer than 317 amino acids, the region from 317 amino acids onwards is not important for the interaction. Thus, we anticipated that one or several amino acids in the cytoplasmic region between 272 and 317 amino acids would be crucial for the interaction. We noticed two characteristic negatively charged residues, E285 and E293, and three positively charged residues, R287, K289, and R291, some of which might form an electrostatic interaction with the 7-TM domain (Fig. 3a and Supplementary Fig. 1). Thus, we replaced each residue with alanine in the KnChR-397 amino acids variant which exhibits fast kinetics, similar to the longest variant (697 amino acids). As shown in Fig. 3b–d, photocurrent decay after switching off light apparently became slower after mutating R287A and R291A. The decay was further slowed in the double mutant R287A/R291A (Fig. 3e). Figure 3f summarizes the time constants of all the mutants tested. The R287A mutant shows a τ-off of 50 ms which is fivefold slower than that of the KnChR-397 variant, while R291A was slowed down to 25 ms. The τ-off of the double mutant R287A/R291A reached about 100 ms which is more than 10 times slower than KnChR-397 amino acids (no amino acid substitution) and is close to the 272 amino acids (no amino acid substitution) variant (130 ms). The three remaining mutants (E285A, K289A, and E293A) showed no effect on kinetics. These results strongly suggest that the positively charged residue, R287 and R291,

interact with the 7-TM domain and contribute to altered channel kinetics (Fig. 3i). The two single mutants (R287 and R291) and the double mutant (R287A/R291A) exhibited significantly larger photocurrents than KnChR-397 (Fig. 3g) without any change in their reversal potential (Fig. 3h).

**Action spectrum and light sensitivity of KnChR.** The action spectrum, a wavelength dependency of photocurrent amplitude, was measured with a multicolor light source. Figure 4 shows that the spectrum of KnCRh-272 photocurrents exhibits the maximum at ~460 nm and a shoulder at ~430 nm, whereas that of KnChR-397 has the maximum at ~440 nm and a shoulder at ~460 nm. It appears that KnChR consists of two isoforms with different absorption maxima. The truncation changes the relative contributions of the two spectral forms. Both absorptions are shorter than that of CrChR2 ($\lambda_{max}$ = 470 nm). To our knowledge, the $\lambda_{max}$ of 430~440 nm is one of the shortest among naturally occurring ChRs.

We then measured the light-power dependency of two KnChR variants (272 amino acids and 397 amino acids), comparing them to CrChR2 (Fig. 5). Photocurrent amplitude, including $I_p$ (peak component) and $I_s$ (steady-state component), grew as light intensity increased. The $I_s$ of the 272 variant became saturated at about 0.09 mW/mm² (Fig. 5a empty circle), while Ip increased further and reached a plateau at about 0.2~0.3 mW/mm² (Fig. 5a filled circle). On the other hand, the light sensitivity of the 397 amino acids variant was markedly lower than that of the 272 amino acids variant. Both $I_p$ and $I_s$ became saturated at about 3–4 mW/mm² (Fig. 5b). Note that the x-axis of each panel differs by a single order of magnitude. The light-power dependency of CrChR2 falls in the same range as KnChR 397 amino acids (Fig. 5c). The half-saturation light intensity ($EC_{50}$) was determined to be 0.051 mW/mm² (KnChR-272), 0.712 mW/mm² (KnChR-397) and 0.731 mW/mm² (CrChR2). This indicates that the C-terminal length affected the light sensitivity of KnChR.

**KnChR exhibits permeability for $H^+$, monovalent and divalent cations.** We predicted that KnChR functions as a cation channelrhodopsin considering its sequence similarity to chlorophyte channelrhodopsins (e.g., CrChR2). Ion selectivity was tested by patch-clamp recording. We used the KnChR 272 amino acids variant for all experiments. Sodium in the extracellular solution was systematically replaced with various cations, while the intracellular solution was fixed (refer to Supplementary Table 1 for details). Figure 6a–f show representative photocurrent traces under various ionic conditions. In the presence of NMG, we assumed that $H^+$ was the conducting ion. The I-V plot under several ionic conditions is depicted in Fig. 6g (KnChR) and Fig. 6h (CrChR2). Data under the rest of the ionic conditions are shown in Supplementary Fig. 7. Note that the peak component (Ip) of the photocurrent was plotted for all the relevant measurements. When Fig. 6a, b are compared, the voltage dependency of the current direction was apparently different. The reversal potential was shifted from −30 to +20 mV (Fig. 6g red and blue), indicating $H^+$ conductance across KnChR (Fig. 6g, i, j, and k). Large $Na^+$ conductance was observed, as shown in Fig. 6c, with a reversal potential of +20 mV (Fig. 6g green, i, j, and k). The shift in the reversal potential of the $K^+$ solution indicates that KnChR conducts $K^+$ (Fig. 6i, j, and k, Supplementary Figs. 7 and 8). We then tested divalent cations, $Ca^{2+}$ and $Mg^{2+}$. We observed an inward photocurrent in the $Ca^{2+}$ solution (Figs. 6e and i) and the reversal potential shifted significantly from −30 mV (NMG pH 9.0) to −5 mV ($Ca^{2+}$ pH 9.0) (Fig. 6g red and black, j, k). A similar observation was observed in the $Mg^{2+}$ solution (Fig. 6f, i, j

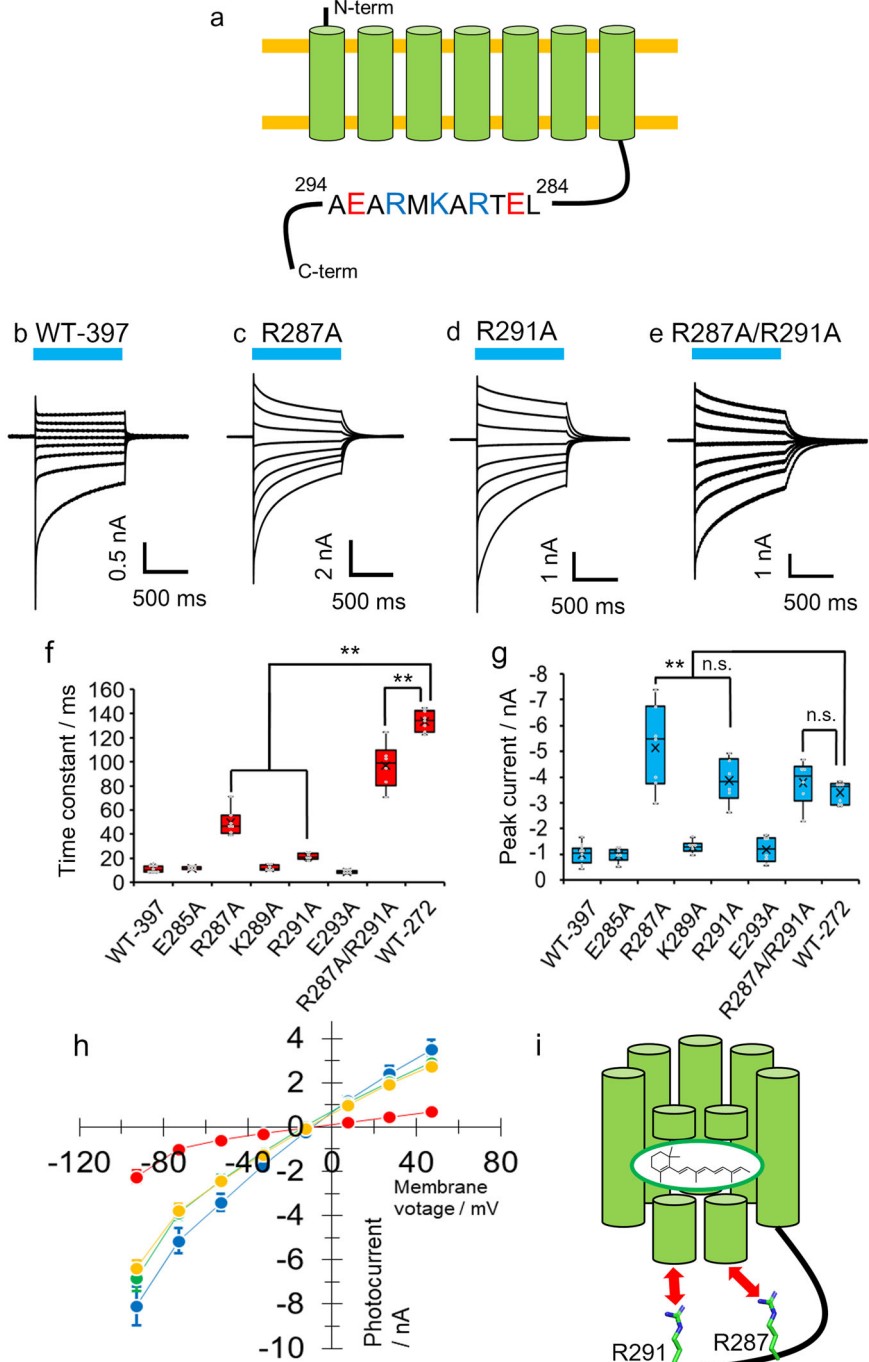

**Fig. 3 Electrophysiological measurements of variants in the C-terminal region. a** Schematic drawing of KnChR with amino acid sequence between 284 and 294. **b–e** Photocurrent traces of KnChR-397 (no mutation), mutant variants- R287A, -R291A, and -R287A/R291A respectively. Blue bars indicate light application (480 nm, 12.3 mW/mm²). Membrane voltage was clamped from at −90 to +50 mV by 20 mV steps. Standard solutions were used (See "Methods"). **f, g** Comparison of time constants and peak current amplitude. The membrane voltage was clamped at −70 mV. $N = 9$ (WT-397), 6 (E285A), 7 (R287A), 6 (K289A), 6 (R291A), 6 (E293A), 6 (R287A/R291A), and 7 (WT-272). **h** I-V relationship of KnChR-397 (red)($N = 9$), -R287A (blue)($N = 7$), -R291A (green)($N = 6$), and –R287A/R291A (yellow)($N = 6$). Data were presented as the mean ± SEM. **i** Schematic drawing of interaction between the 7-TM domain and the cytoplasmic region in KnChR.

and k, Supplementary Figs. 7 and 8). Altogether, we demonstrated that KnChR is a cation channelrhodopsin that exhibits permeability for monovalent and divalent cations. Figure 6h shows the I-V plot of Cr_ChR2 under the same condition as Fig. 6g. Reversal potentials and a shift in reversal potential (ΔE_rev) of KnChR and CrChR2 are depicted in Figs. 6j, k, and l, respectively. The relative permeability depends not only on the

$\Delta E_{rev}$ measured after substituting the test ion for the control ion, but also on the actual concentrations of the two ions. Taking into account ion concentrations used for the measurements, we estimated permeability ratios (Table 1). KnChR actually exhibited far greater relative permeability for $H^+$ than for the metal cations, as is typical of chlorophyte CCRs such as CrChR2. For example, the KnChR permeability ratio of $Na^+$ to $H^+$ ($P_{Na+}/P_{H+}$) was

estimated as $12 \times 10^{-7}$, while that of CrChR2 was $1.1 \times 10^{-7}$, indicating KnChR is ~10-fold higher permeable for $Na^+$. The permeability ratio of $K^+$ to $H^+$ ($P_{K+}/P_{H+}$) of KnChR is about 14-fold higher than that of CrChR2 ($8.7 \times 10^{-7}$ and $0.63 \times 10^{-7}$ respectively). Interestingly, the KnChR permeability ratio of $Ca^{2+}$ ($P_{Ca2+}/P_{Na+}$) is higher (0.25) than that of CrChR2 (0.15).

As mentioned above, we analyzed the peak component of each photocurrent ($I_p$). In addition, the steady-state component ($I_s$) were analyzed to estimate $E_{rev}$ and $\Delta E_{rev}$. Essentially the same results were found (Supplementary Fig. 8).

## Discussion

In this study, we report on the electrophysiological characterization of a newly identified multi-domain cation channelrhodopsin from *Klebsormidium nitens* (KnChR). KnChR possesses a large cytoplasmic domain made of about 540 amino acids which might involve a peptidoglycan binding moiety (FimV). For the first time, we demonstrated that the C-terminus of KnChR affects the light-gated ion channel function. The channel closure rate (τ-off) slowed down from 10 to 130 ms as the cytoplasmic domain was shortened (Fig. 2h). In other words, the C-terminus domain accelerates channel closure. Positively charged residues, R287 and R291, in particular, are crucial for the acceleration, as the double mutant R287A/R291A reduced the fast kinetics (Fig. 3f). Furthermore, light sensitivity was altered following systematic truncation of the C-terminus (Fig. 5a and b). Therefore, we propose that an electrostatic interaction exists between the residues of the C-terminus and the 7-TM rhodopsin domain of KnChR (Fig. 3i). Although we did not identify the counterparts of R287 and R291 in this study, the candidates could be negatively charged residues such as Asp and Glu in the TM domain, which would be investigated in the future. Modulation of ion-transport

properties by the cytoplasmic domain has never been proven in ion-transporting rhodopsin thus far. However, the previous study revealed that the cytoplasmic domain controls the intramolecular charge movements in Anabaena sensory rhodopsin[20]. In the case of enzyme rhodopsin (HKR or 2C-cyclops), the full-length HKR has been shown to exhibit a normal photocycle which reverts to the initial dark state in 20 s, whereas truncation of the enzyme domain at the C-terminus region resulted in the production of a bi-stable photocycle in which rhodopsin possesses two dark states (400 and 380 nm), which exhibit photochromism[21,22]. Detailed functional characterization of KnChR with spectroscopy and a structural study together with mutation analysis would be needed to reveal the mechanism of the functional modulation by the cytoplasmic domain.

The action spectrum of KnChR, assessed from a voltage clamp experiment, revealed its absorption maxima ($\lambda_{max}$) at 430 and 460 nm depending on the protein length (Fig. 4). To the best of our knowledge, this is one of the most blue-shifted $\lambda_{max}$ in naturally occurring ChRs. Govorunova et al. previously reported a blue-shifted ChR from *Platymonias subcordiformus* (PsChR) that displayed an action spectrum of $\lambda_{max} = 445$ nm, and an absorption maximum of 437 nm after purification (in solution)[23]. They ascribed the blue-shift to amino acid residues G142 and A146 in 4-TM (T159 and G163 in CrChR2). The replacement of these residues resulted in a blue shift when tested with C1C2 and iC+ +[24,25]. KnChR conserves A146 (as A161) while G142 is replaced by A157. Thus, A161 would be one of the color determinants in KnChR.

The value of a blue-shifted ChR for use in optogenetics was put forth when PsChR was reported in 2013[23]. This idea was recently confirmed and extended by Duan and coworkers[26].

The comparison of permeability ratio indicates that KnChR is highly $Na^+$ conductive compared to CrChR2 (~10 fold higher than CrChR2) (Table 1). Previous reports revealed that PsChR shows a high $Na^+$ selectivity, in which $P_{Na+}/P_{H+}$ of PsChR is about 6-fold higher than that of CrChR2[23,26]. Thus, $Na^+$ selectivity of KnChR is in the same range as that of PsChR. The PsChR D139H mutant exhibited the further increased $P_{Na+}/P_{H+}$ up to 30-fold[26]. Mutation study for further increasing $Na^+$ selectivity of KnChR would be needed for optogenetics application because high $Na^+$ conductive ChR is more suitable as a depolarization tool. KnChR conducts $Ca^{2+}$ at a relatively high amount ($P_{Ca2+}/P_{Na+} = 0.25$) (Table 1) which is in the same level of that of an engineered $Ca^{2+}$-conductive variant so-called Catch (CrChR2 L132C) ($P_{Ca2+}/P_{Na+} = 0.24$)[27]. As PsChR D139H exhibited enhanced $Ca^{2+}$ conductance, it has a great potential for engineering even higher $Ca^{2+}$ permeable KnChR[26].

As for the physiological role of the cytoplasmic domain, phosphorylation site prediction of KnChR suggested that several motifs may be involved in phosphorylation (Supplementary Fig. 9). Three phosphorylation sites for CrChR1 and one for

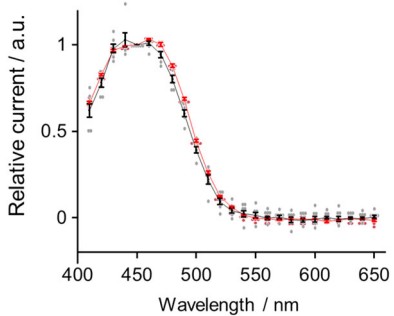

**Fig. 4 Action spectrum of KnChR-272 and KnChR97.** Wavelength dependency of the photocurrent. The photocurrents were measured in standard solutions. The membrane voltage was clamped at −40 mV. Red, KnChR-272 ($N = 5$), Black, KnChR-397 ($N = 6$). Data were presented as the mean ± SEM.

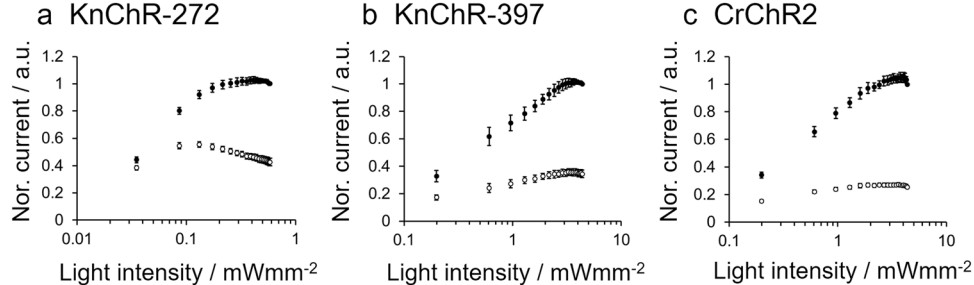

**Fig. 5 Light intensity dependence of KnChR-272, KnChR-397, and CrChR2.** The photocurrent was measured with various intensities of 480 nm light. **a** KnChR-272, **b** KnChR-397 and **c** CrChR2. Membrane voltage was clamped at −60 mV. Standard solutions were used (See "Methods"). Black circle, peak current; empty circle, steady-state current. Data were presented as the mean ± SEM. $N = 6$ (**a**), 4 (**b**), and 6 (**c**).

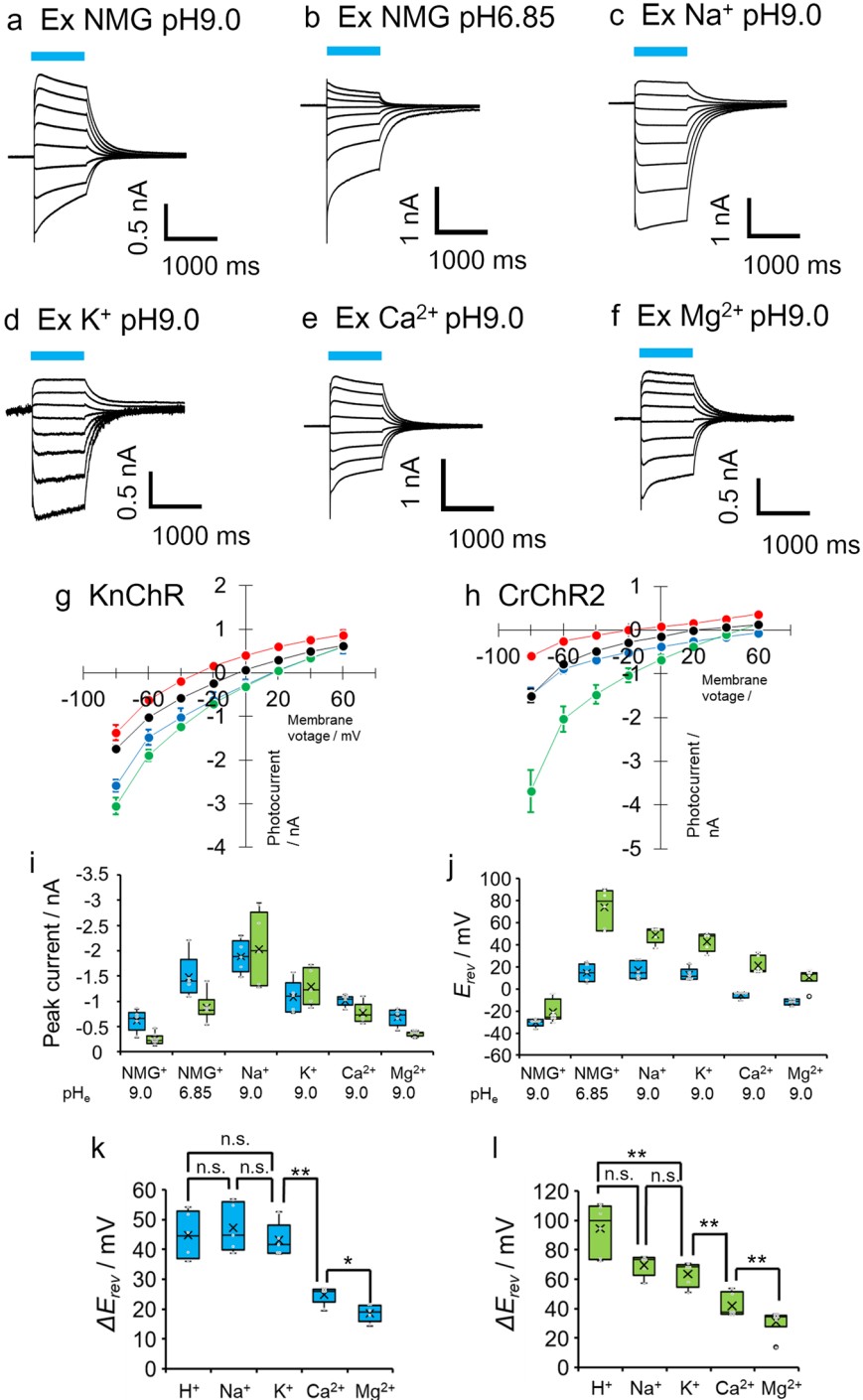

**Fig. 6 Ion selectivity of KnChR and CrChR2. a–f** Representative photocurrent traces of KnChR-272 under various cation conditions. Solutions were listed in Supplementary Table 1. Blue bars indicate light application (480 nm, 12.3 mW/mm²). Membrane voltage was clamped from at −90 to +50 mV by 20 mV steps. I-V relationship of KnChR (**g**) and CrChR2 (**h**). The peak component of photocurrent was plotted. Red: Ex NMG pH 9.0, Blue: Ex NMG pH 6.85, Green: Ex Na+ pH 9.0, Black: Ex Ca²+ pH 9.0. **i** Comparison of peak current amplitude under various ionic conditions at −60 mV. light blue, KnChR; green CrChR2. **j** Reversal potential $E_{rev}$ under various solutions. **k** Reversal potential shift ($\Delta E_{rev}$) of KnChR. $\Delta E_{rev}$ of each cation was calculated by subtracting $E_{rev}$ of each ionic condition in (**j**) by $E_{rev}$ of NMG+ at pH$_e$ 9.0. e.g., $\Delta E_{rev}$ of H+ is determined as $E_{rev}$ (NMG+ pH$_e$ 6.85) - $E_{rev}$ (NMG+ pH$_e$ 9.0). **l** Reversal potential shift ($\Delta E_{rev}$) of CrChR2. Data were presented as the mean ± SEM. $N = 5$ (all the conditions for KnChR). $N = 8$ (CrChR2 in NMG pH 9.0), 6 (CrChR2 in NMG pH 6.85), 5 (CrChR2 in Na+ pH 9.0), 5 (CrChR2 in K+ pH 9.0), 6 (CrChR2 in Ca²+ pH 9.0), and 6 (CrChR2 in Mg²+ pH 9.0).

CrChR2 exist, suggesting a kinase-dependent regulation of function such as photomotility and calcium signaling of green alga[15]. It would be interesting to delineate the functional role of these predicted phosphorylation patterns of KnChR in a native system.

Moreover, the presence of a FimV domain of KnChR at the cytoplasmic side implies that it might interact with the peptidoglycan layer. Previous studies suggested the presence of peptidoglycan layer around the chloroplast in glaucophyte and streptophyte (specifically in *Klebsormidium nitens*) of algal

**Table 1 Permeability ratios.**

|  | $Na^+/H^+$ | $K^+/H^+$ | $Na^+/K^+$ | $Ca^{2+}/Na^+$ |
|---|---|---|---|---|
| CrChR2 | $1.1 \times 10^{-7}$ | $0.63 \times 10^{-7}$ | 1.9 | 0.15 |
| KnChR | $12 \times 10^{-7}$ | $8.7 \times 10^{-7}$ | 1.5 | 0.25 |

The permeability ratio for each cation was estimated by the Goldmann–Hodgkin–Katz equation[33].

groups but not in chlorophyte group[28]. This algal peptidoglycan layer is similar to peptidoglycan present in bacterial cell wall since it was sensitive to antibiotic (penicillin) and D-cycloserine (inhibitor of D-Ala: D-Ala ligase). The peptidoglycan layer serves as an essential system for chloroplast division in *K. nitens*[29]. Blocking of peptidoglycan generation arrested cell division and inhibited chloroplast division. Also, the presence of peptidoglycan biosynthetic pathway genes such as FtsZ3 (Filamentous temperature-sensitive Z) involved in chloroplast maintenance and division in Streptophyte strengthens the presence of FimV domain in this system (*K. nitens*)[30,31]. Thus, the presence of FimV domain with a light sensor is unique and its role via light input is an open question. Interaction of the FimV of KnChR with peptidoglycan and its effect needs further experimental evidence.

## Methods

**Expression plasmids.** Synthesized genes encoding KnChR and CrChR2 (1–315 amino acids) were subcloned into a phKR2-3.0-eYFP plasmid (gifted by Dr. H. Yawo (University of Tokyo, Japan)) using an In-Fusion HD cloning kit (Takara Bio, Shiga, Japan). Length variants of KnChR (amino acid length of 272, 280, 290, 300, 310, 317, 321, 397, 697, and 831) were created by using appropriate PCR primers. For the immunostaining experiment, N-QKLISEEDL-C (10 amino acids, c-Myc epitope tag) in the C-terminal of eYFP was inserted in the plasmid pKnChR (697 amino acids)-3.0-eYFP using inverse PCR. Site-directed mutagenesis was performed using a QuikChange site-directed mutagenesis kit (Agilent, CA, USA). All the constructs were verified by DNA sequencing (Fasmac Co., Ltd. Kanagawa, Japan). All the PCR primers used in this study were summarized in Supplementary Tables 2–4.

**Mammalian cell culture.** Electrophysiological assays and immunostaining of KnChR and CrChR2 were performed on ND7/23 cells, which are hybrid cell lines derived from neonatal rat dorsal root ganglia neurons fused with mouse neuroblastoma. ND7/23 cells were grown on a coverslip in Dulbecco's modified Eagle's medium (DMEM; Wako, Osaka, Japan) supplemented with 2.0 μM of all-*trans* retinal and 5% fetal bovine serum, and under a 5% $CO_2$ atmosphere at 37 °C. The expression plasmids were transiently transfected by using the FuGENE® HD Transfection Reagent (Promega, Fitchburg, WI, USA) according to the manufacturer's instructions. Electrophysiological recordings were then conducted 24–36 h after transfection. Successfully transfected cells were identified by eYFP fluorescence under a microscope prior to measurements.

**Immunostaining of mammalian cells.** ND7/23 cells were cultured on glass coverslips. Cells expressing KnChR or KnChR bearing the c-Myc epitope tag at the C-terminus were fixed in 4% paraformaldehyde phosphate buffer solution for 15 min at room temperature. Cells were then washed three times with phosphate-buffered saline (PBS). When necessary, cells were permeabilized with 0.5% Triton X-100 for 15 min at room temperature. Cells were treated with a blocking buffer consisting of 3% goat serum for 60 min at room temperature. After blocking, the cells were incubated with rabbit anti–c-Myc primary antibody (C3956; Sigma-Aldrich, St. Louis, MO, USA) at a 1:500 dilution for 60 min at room temperature, then washed three times with PBS and labeled with goat anti-rabbit IgG secondary antibody Alexa Fluor 594 (A-11037; Thermo Fisher Scientific, Waltham, MA, USA) at a 1:200 dilution for 2 h at room temperature. After a final wash with PBS, the coverslips were mounted on glass slides with ProLong Diamond Antifade Mountant (Thermo Fisher Scientific).

Live cultured KnChR bearing the c-Myc epitope tag at the C-terminus expressing-cells were washed with PBS. The cells were moved from the culture medium to serum-free DMEM and incubated with rabbit anti–c-Myc primary antibody at a 1:500 dilution for 60 min under a 5% $CO_2$ atmosphere at 37 °C. The cells were washed two times with PBS before fixing with 4% paraformaldehyde phosphate buffer solution for 15 min at room temperature. The cells were washed three times with PBS before labeling with goat anti-rabbit IgG secondary antibody Alexa Fluor 594 at a 1:200 dilution for 2 h at room temperature. After a final wash

with PBS, the coverslips were mounted on glass slides with ProLong Diamond Antifade Mountant.

Fluorescent images were acquired with fluorescent microscopy LSM880 equipped with ×40 objective lens (Zeiss, Jena, Germany) and a software ZEN (Zeiss). The captured images were analyzed with Fiji software[32].

**Electrophysiology.** All experiments were carried out at room temperature (22 ± 2 °C). Photocurrents were recorded with an Axopatch 200B amplifier (Molecular Devices, Sunnyvale, CA, USA) under a whole-cell patch-clamp configuration[7]. Data were filtered at 5 kHz and sampled at 20 kHz (Digdata1550, Molecular Devices, San Jose, CA, USA) and stored in a computer (pClamp10.7, Molecular Devices). Pipette resistance was 3–6 MΩ. The standard internal pipette solution for the whole-cell voltage-clamp contained (in mM) 126 NaAsp or NaCl, 0.5 $CaCl_2$, 2 $MgCl_2$, 5 EGTA, 25 HEPES, 12.2 NMG, and adjusted to pH 7.4 by citric acid. The standard extracellular solution for the whole-cell voltage-clamp contained (in mM) 150 NaCl, 1.8 $CaCl_2$, 1 $MgCl_2$, and 10 HEPES, 10 NMG, 5 glucose, and adjusted to pH 7.4 by NMG. Internal and external solutions for ion selectivity are shown in Supplementary Table 1. The liquid junction potential was calculated and compensated by pClamp 10.7 software. Data were analyzed by Clampfit 10.7 software (Molecular Devices). Time constants were determined by a single exponential fit. The half-saturation light intensity ($EC_{50}$) was determined by using Igor pro 4.06 software (Hulinks, Tokyo, Japan). The permeability ratio for each cation was estimated by the Goldmann–Hodgkin–Katz equation[33].

**Optics.** For whole-cell voltage-clamp, irradiation at 480 nm was carried out using collimated LED (parts No. LCS-0470-03-22, Mightex, Toronto, Canada) controlled by computer software (pCLAMP10.7, Molecular Devices). Light power was measured directly by an objective lens of a microscope by a power meter (LP1, Sanwa Electric Instruments Co., Ltd., Tokyo, Japan). All action spectra were measured at the same light intensity in the range of 410–650 nm by a xenon light source OSG (Hamamatsu photonics, Hamamatsu, Japan).

**Statistics and reproducibility.** Data were presented as the mean ± SEM of all experiments with $N$ = number of biological replicates. The box-and-whisker plots represent the median (center line), the mean (x), interquartile range (box limits) and 1.5 × interquartile range (whiskers). Data were evaluated with the Mann–Whitney $U$ test for statistical significance unless noted otherwise. Means were judged as statistically insignificant when $P > 0.05$.

**Reporting summary.** Further information on research design is available in the Nature Research Reporting Summary linked to this article.

## Data availability

Data supporting the findings of this manuscript have been deposited in Figshare.com (https://figshare.com/s/5c04d301c1a450621980).

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

## Acknowledgements

We thank Miki Iwatani, Ryoko Nakamura, and Kyoko Tsunoda for their excellent technical assistance. This work was financially supported by the Japanese Ministry of Education, Culture, Sports, Science and Technology (25104009, 15H02391 to H.K. and 18K06109 to S.P.T.), a JST CREST grant (JPMJCR1753 to H.K.), and a JST PRESTO grant (JPMJPR1688 to S.P.T.). SK is thankful to the Department of Biotechnology, Government of India for providing an IYBA project-BT 101 O/IYBAJ20 16/02.

## Author contributions

S.K., S.P.T., and H.K. devised the initial idea for the project. K.S. and S.S. performed bioinformatic analyses. S.H. performed immunostaining experiments. R.T. performed electrophysiological experiments. S.P.T. prepared the manuscript with contributions from all of the authors to the data analysis, figure generation, and the final manuscript.

## Competing interests

The authors declare no competing interests.
