## [Peer Review File · Communications Biology]

Reviewers' comments:

Reviewer #1 (Remarks to the Author):

The manuscript by Tashiro et al. entitled "Unique light-gated ion channel properties of a novel modular cation channelrhodopsin from an evolutionary important terrestrial green alga" reports identification and characterization of a new channelrhodopsin variant and demonstration that its channel properties (namely, the kinetics of photocurrent decay) are influenced by specific residues in the cytoplasmic fragment of the polypeptide outside of the transmembrane (rhodopsin) domain. To the best of my knowledge, the latter has not yet been reported in any other channelrhodopsin and is therefore an important observation. Although there is little more in this manuscript, I think a revised version can be published in *Communications Biology* provided the Authors take into account the following issues.

Major issues:

Lines 3-5: "Unique light-gated ion channel properties of a novel modular cation channelrhodopsin from an evolutionary important terrestrial green alga"

I find this title highly misleading. The Authors call KnRh3 a "modular protein" because it comprises a predicted peptidoglycan binding domain (FimV; residues 410-690) in addition to the rhodopsin domain. However, the residues that they found to influence the channel kinetics (287 and 291) are NOT located within FimV. Moreover, the decay rates in the constructs with and without FimV (comprising 697 and 397 residues, respectively) were not significantly different (Fig. 2H). Therefore, FimV does not appear to be important for channel activity, as the Authors themselves conclude (Lines 271-272). Also, such words as "unique" and "novel" are not informative. I suggest the Authors choose a different, more to the point title, e.g. "Specific residues in the cytoplasmic domain modulate photocurrent kinetics of channelrhodopsin from the alga *Klebsormidium nitens*".

Lines 256-261: "We reasoned that current amplitude depends on the open-life time of the channels... M-state accumulated during illumination"

While it is true that the current amplitude recorded in response to pulses of continuous light depends on the open time of the channel, it also depends on the number of functional molecules in the membrane. (Note that this number cannot be assessed by simply measuring the tag fluorescence, as these measurements do not report the functional state of the channel). Therefore, the conclusions drawn in this paragraph appear to be pure speculation. To justify these conclusions, the Authors would need to purify the protein and probe its photocycle by flash photolysis. In the absence of such data, they should delete this speculation. Longer constructs are generally known to show lower expression levels, and indeed, the Authors observed only poor expression of the full-length construct (Lines 212-213). Therefore, poorer expression seems to be the most likely reason for smaller currents recorded from the longer constructs as compared with the shorter ones.

Lines 267-268: "The I-V plots of KnRh3 show a rather weak rectification and relatively large outward currents were observed (see +50 mV in Fig. 2J and K)."

Rectification is defined as the ratio of the currents measured at the positive and negative voltages of the same absolute magnitude. Indeed, Figures 2J and K show larger outward currents from KnRh3 than from CrChR2, but inward currents from the former are also larger than those from the latter, so the two channelrhodopsins appear to have similar degrees of inward rectification. The same conclusion can be drawn from comparison of panels G and H in Figure 7.

Lines 357-358: "showed no significant difference in ΔE_{rev} among H⁺, Na⁺ and K⁺ (45~48 mV), indicating that permeability of these cations was almost the same (Fig. 7K)."

It is not clear what the Authors mean by "permeability" here. The permeability RATIO (relative permeability) can be derived using the Goldman-Hodgkin-Katz equation. The relative permeability depends not only on the ΔE_{rev} measured after substituting the test ion for the control ion, but also

on the actual concentrations (activities) of the two ions (see e.g. [PMID: 1431803]). Taking into account that the Authors used ~1400000-fold higher concentrations of Na⁺ or K⁺ than of H⁺ in their bath solutions, KnRh3 actually exhibited far greater relative permeability for H⁺ than for the metal cations, as is typical of chlorophyte CCRs (see e.g. [PMID: 14615590]).

Minor issues:

Line 23: "novel channelrhodopsin KnRh3"

The names of all so far identified channelrhodopsins have been abbreviated as ChRs or, more recently, as CCRs or ACRs for cation and anion channelrhodopsins, respectively. On the other hand, the abbreviation "Rh" is usually reserved for animal visual rhodopsins that form a different superfamily, so the Authors' using it here will unavoidably confuse the readers. As the Authors have clearly demonstrated cation channel activity of this protein, I suggest they use the abbreviations "KnCCR" or "KnChR" instead of KnRh3. I understand that the Authors would like to keep the protein name consistent with their earlier preprint (<https://www.preprints.org/manuscript/202009.0015/v1>), but they could simply explain here that KnRh3 is a synonym.

Lines 49-58: "Microbial-type rhodopsins... and other ion-pumping rhodopsins."

A general description of the entire superfamily of microbial rhodopsins is not relevant for this study, as KnRh3 is highly homologous to the known CCRs from green algae (Lines 186-187). Please delete it. Instead, background information on the cytoplasmic domains in channelrhodopsins would be useful to introduce the readers to the problem. To this end, please move the lines 366-378 from Discussion to Introduction.

Line 196: "His instead of Asp at position 96 was regarded as a characteristic of channelrhodopsins"

Please delete this statement as grossly outdated. This conclusion has been drawn from comparison of the four closely related ChRs from *Chlamydomonas* and *Volvox* that were identified first; since then, hundreds of new ChRs were discovered in which this residue is not conserved.

Lines 197-198: "this position was occupied by Ala, by D156 in GtACR1, while C128 in CrChR2 formed a hydrogen bridge (D-C pair or D-C gate) which altered the channel open lifetime"

In GtACR1, the homolog of C128 is C102, not D156, so it is unclear what the Authors wanted to say in this sentence. Moreover, it is unclear why they mention GtACR1 in the first place. Please clarify.

Lines 230-231: "As shown in Fig 2 B-F, large photocurrents were observed in all the five KnRh3 variants"

Please delete the word "large", because this statement contradicts that in Line 253: "the photocurrent amplitudes also differed among the nine variants (Fig. 2I)." The latter figure shows that indeed the currents from the variants comprising >300 residues were significantly smaller than those from the shorter constructs.

Line 278: "compare the channel-off kinetics and amplitude (Fig. 3)"

I suggest to move Figure 3 to supplement because it actually shows control data (verification that the observed differences in the current kinetics are independent on the presence of the fluorescent tag).

Line 283: "In fact, eYFP constructs contain the ER-exporting signal peptide"

This should be explained from the beginning, i.e. in Line 211, to avoid confusing the readers.

Line 303: "The double mutant R287A/R291A reached about 100 ms"

It was not the mutant itself, but the t-off of its current decay that reached this value; please correct.

Line 315: "spectrum with two λ_{max} at 430 and 460 nm"

By definition, the maximum is the largest value, so there can only be one maximum of the spectrum. Figure 5 shows that the spectrum of KnRh3-272 photocurrents exhibits the maximum at ~460 nm and a shoulder at ~440 nm, whereas that of KnRh3-397 has the maximum at ~440 nm and a shoulder at ~460 nm. So, it appears that the truncation changes relative contributions of the two spectral forms, which seems to be worth discussing in the text.

Line 331: "KnRh3 permeates monovalent and divalent cations"

Cations permeate through the channel, not the other way around; please correct.

Line 366: "Several ChRs also have a long C-terminus tail"

"Several" is an understatement here. To the best of my knowledge, ALL so far known channelrhodopsins comprise a few hundreds of amino acid residues in their cytoplasmic domains.

Lines 393-394: "Modulation of photocycle kinetics by the cytoplasmic domain has never been proven in ion-transporting rhodopsin thus far"

While this is correct, perhaps the Authors would like to cite the study in which it was demonstrated that the cytoplasmic domain controls the intramolecular charge movements in Anabaena sensory rhodopsin [PMID: 17012323].

Lines 418-419: "The cation channelrhodopsin from Cryptophyte algae (DTD channelrhodopsin) exhibited different ratio of cation permeation"

This statement makes an impression that the higher permeability for metal cations relative to proton distinguishes cryptophyte CCRs from their chrolophyte counterparts. This is however not true: for example, PsChR from Ref. 19 showed a similar Na^+/H^+ permeability ratio to that of cryptophyte BCCRs.

Table 1: For the bath solution named "NMG6.85" no NMG is listed among the ions. Do I assume correctly that it was 140 mM NMG, and the pH was adjusted with MES? If so, please correct.

Reviewer #2 (Remarks to the Author):

This manuscript identifies and characterizes a new cation channelrhodopsin(KnRh3)from a terrestrial algae (Klebsormidium nitens). The channelrhodopsins have found important application in the burgeoning field of optogenetics, though this particular example may not prove to be useful in this application due to its short wave (430-460nm) absorption, one of the most blue shifted channelrhodopsins characterized thus far. This is also one of, if not the first channelrhodopsin from a terrestrial algae to be characterized. The authors characterized the channel using patch clamp to measure ion current and kinetics of the channel. The authors showKnRh3 to be a relatively nonselective cation channel, showing channel activity for a variety of cations including H^+ , Na^+ , K^+ and even divalent cations such as Ca^{2+} . The most interesting characteristics of this new channel include it's blue shifted spectrum and the fact that the c-terminal domain regulates the channel, which has not been previously observed in a channelrhodopsin ion channel. The kinetics of channel closure is modulated from 10 ms to over 100ms by the c-terminal domain. This is shown first by deletion analysis, with the effect then localized to two basic residues in the domain. They also found that truncation of the c-terminal domain also lead to substantially increased light sensitivity(line 318-329). It would be interesting to know if the light sensitivity is also effected by the two above mentioned residues. This point should be addressed in the manuscript.

Overall, the data looks to be well-measured and analyzed and supports the conclusions of the paper. The findings are novel and of interest to both the rhodopsin field and to a broader field of researchers interested in algal adaptation to terrestrial existence, a key step in the evolution of land plants.

Line 32: ...two Arginine residues, R287 and R291, that are crucial...

Line 90: FimV is involved in peptidoglycan binding protein...(this sentence needs to be clarified)

Line 95:light sensitivity and ion selectivity were compared...Line 148: ...fluorescent images were acquired with a fluorescent microscope...

Line 241...this indicates that the channel kinetics and the photocycle are effected by the cytoplasmic domain...The authors should elaborate more to explain exactly how the observations demonstrate a change in the photocycle.

Line 361: ...in addition,Is was analyzed.... (the meaning here is not clear and should be rewritten)

Line 379: should read: Phosphorylation site prediction of KnRh3 suggested that several motifs may be involved in phosphorylation

Reviewer #3 (Remarks to the Author):

By genome screening the authors have identified a channelrhodopsin (ChR) called KnRh3 from an alga different in habitat (terrestrial) and morphology (filamentous) from those previously reported. There are three different properties of KnRh3 that the authors identify in their manuscript:

(1) A unique structural feature is that following the membrane-inserted opsin domain, the ChR contains an FimV domain in the cytoplasmic C-terminal region. FimV is a peptidoglycan-binding protein found in gram-negative bacteria and has been shown to be involved in pilus assembly, migration of components of cells involved in cell-division, and motility, all of which are likely to involve FimV binding to the peptidoglycan layer found in gram-negative bacteria. Since algae do not have a peptidoglycan layer in their cell wall, the role of FimV and its association with a light-activated cation channel is mysterious and interesting.

(2) KnRh3 is only the second of the far-blue-shifted ChRs known, the first, PsChR, reported in a marine alga in 2013. This property is of interest to users of ChRs as optogenetic tools.

(3) The authors demonstrate that the cytoplasmic region interacts with the 7-TM opsin domain and slows channel closing, and present mutant data indicating that electrostatic interaction between the opsin domain and the cytoplasmic domain accelerates channel closing. No such effect was observed in measurements with opsin domain vs full-length CrChR2.

I believe that there is sufficient potential importance of KnRh3 to eventually publish an article on its properties listed above. The work presented in this submission provides a good start, but is in several respects preliminary. Before publication, the authors need to bring the level of understanding of the molecule, especially of aspects (1) and (2), beyond a brief scanning of properties followed by speculative ideas. I list comments regarding some concerns and identify additional discussion and results that could provide further knowledge about KnRh3 to complete the manuscript.

Comments:

1. The authors state several proposals and conclusions about the photocycle, such as "electrostatic interaction between the 7-TM domain and the C-terminal domain accelerates the photocycle" and "Thus, the apparent current amplitude of the short variants was elevated as the M-state accumulated during illumination". Since the authors do not report any photocycle measurements in the submission, any comments regarding intermediates and their lifetimes are speculative and

unjustified. Flash spectroscopy of the pigment would relate the photocycle to the ion currents.

2. The authors claim that "In addition, KnRh3 would expand the optogenetics tool kit, especially for when short wavelength excitation is required." The value of a blue-shifted ChR for use in optogenetics was put forth when the first blue-shifted one, PsChR, was reported in 2013 (authors' reference 19). This idea was confirmed and extended recently by Nagel and coworkers (Duan X, Nagel G, and Gao S. 2019. Mutated Channelrhodopsins with Increased Sodium and Calcium Permeability. *Appl. Sci.* 2019, 9(4), 664; <https://doi.org/10.3390/app9040664>). The authors need to describe prior published studies in terms of relevance to optogenetics and compare KnRh3 properties with those of PsChR. If the authors want to claim that the blue-shifted KnRh3 would be useful as an optogenetics tool, they need to compare the efficacy of neuron activation with that previously published for the similarly blue-shifted PsChR. Are there advantages to KnRh3 over PsChR for optogenetics use? The authors compare properties of KnRh3 important for optogenetic tools, such as ion selectivity, with CrChR2, but the most relevant comparison would be to PsChR.

3. The action spectrum may be influenced by other pigments with unknown absorption spectra altering the intensity of the actinic light. The shape of the action spectrum with 2 equal extinction peaks needs explanation. Is it due to vibrational fine structure? Two conformations of the pigment? Influence of unrelated pigments in the cells? The authors need to examine the absorption spectrum of the purified KnRh3. In vitro analysis of KnRh3 would also facilitate the flash spectroscopy (Comment 1) and comparison with PsChR needed (Comment 2).

4. An unusual feature of KnRh3 is that it includes an FimV domain, a peptidoglycan found typically in gram-negative bacteria, at the C-terminus of rhodopsin. FimV is a peptidoglycan-binding protein which has been found to be involved in secretion, cell-division, and motility in gram-negative bacteria. The authors conclude that "it is anticipated" that light signals are transduced via ChR into motility in *K. nitens*. This conclusion is not justified. The processes (motility being only one of them) in gram-negative bacteria are likely due to FimV binding to the peptidoglycan layer. Since algae do not have a peptidoglycan layer in their cell wall, the role of FimV is an open, and very interesting, question. Peptidoglycan is found in some algae in their prokaryotic-derived photosynthetic organelles in their cytoplasm. Is peptidoglycan found in *K. nitens*? If so, is it located in the same compartment of the cell as the cytoplasmic domain of KnRh3? Does peptidoglycan bind to KnRh3 protein, and, if so, does it alter its channel properties? To start, the authors need to summarize the literature regarding peptidoglycan in algae.

5. A general problem is that the manuscript is sometimes unclear because the authors use vague language. This problem starts from the beginning in their title. First, "unique light-gated ion channel properties": What are the unique properties? There do not appear to be unique channel properties shown in the article. The aspect of the molecule that may be unique is the presence of a peptidoglycan binding domain (FimV), but no evidence is presented that FimV plays a role in activity of the channel. Second, "novel modular": Essentially all channelrhodopsins are modular with cytoplasmic domains outside of the opsin domain, so the modularity itself is not novel. Third, the authors emphasize that the algal species encoding KnRh3 is an "evolutionary important alga"? The authors do not discuss any evolutionary implications of their findings, as would be expected from this claim being high-lighted in the title. The authors need to read through their manuscript and make sure that each sentence has a well-defined meaning. Such a rewriting would greatly improve the manuscript and help readers understand the authors' interpretation of their data.

Reviewers' comments:

Reviewer #1 (Remarks to the Author):

The manuscript by Tashiro et al. entitled “Unique light-gated ion channel properties of a novel modular cation channelrhodopsin from an evolutionary important terrestrial green alga” reports identification and characterization of a new channelrhodopsin variant and demonstration that its channel properties (namely, the kinetics of photocurrent decay) are influenced by specific residues in the cytoplasmic fragment of the polypeptide outside of the transmembrane (rhodopsin) domain. To the best of my knowledge, the latter has not yet been reported in any other channelrhodopsin and is therefore an important observation. Although there is little more in this manuscript, I think a revised version can be published in *Communications Biology* provided the Authors take into account the following issues.

Major issues:

Lines 3-5: “Unique light-gated ion channel properties of a novel modular cation channelrhodopsin from an evolutionary important terrestrial green alga”

I find this title highly misleading. The Authors call KnRh3 a “modular protein” because it comprises a predicted peptidoglycan binding domain (FimV; residues 410-690) in addition to the rhodopsin domain. However, the residues that they found to influence the channel kinetics (287 and 291) are NOT located within FimV. Moreover, the decay rates in the constructs with and without FimV (comprising 697 and 397 residues, respectively) were not significantly different (Fig. 2H). Therefore, FimV does not appear to be important for channel activity, as the Authors themselves conclude (Lines 271-272). Also, such words as “unique” and “novel” are not informative. I suggest the Authors choose a different, more to the point title, e.g. “Specific residues in the cytoplasmic domain modulate photocurrent kinetics of channelrhodopsin from the alga *Klebsormidium nitens*”.

Response: Thank you for the comment. We named KnRh3 as modular channelrhodopsin, since, no other known channelrhodopsins encode another putative functional domain. It might have light-gated FimV activity. However, as the reviewer mentioned, we conclude that no functional relation of FimV was elucidated in current study. Suggestion for revision of the title is fine and the same is accepted.

Lines 256-261: “We reasoned that current amplitude depends on the open-life time of the channels... M-state accumulated during illumination”

While it is true that the current amplitude recorded in response to pulses of continuous light depends on the open time of the channel, it also depends on the number of functional molecules in the membrane. (Note that this number cannot be assessed by simply measuring the tag fluorescence, as these measurements do not report the functional state of the channel). Therefore, the conclusions drawn in this paragraph appear to be pure speculation. To justify these conclusions, the Authors would need to purify the protein and probe its photocycle by flash photolysis. In the absence of such data, they should delete this speculation. Longer constructs are generally known to show lower expression levels, and indeed, the Authors observed only poor expression of the full-length construct (Lines 212-213). Therefore, poorer expression seems to be the most likely reason for smaller currents recorded from the longer constructs as compared with the shorter ones.

Response: We thank the reviewer for the valuable suggestions. We have actually tried to express and isolate the protein to measure the photocycle by flash photolysis. But no success for obtaining sufficient functional expression of KnChR so far. Thus, it is currently difficult to assess the photocycle. Taken the reviewers words, we delete the conclusion about the relation between current amplitude and the M-intermediate accumulation in the entire text. Besides we describe that the smaller current amplitudes is derived from poorer expression level. Please see Line 238-243 in the revised text.

Lines 267-268: “The I-V plots of KnRh3 show a rather weak rectification and relatively large outward currents were observed (see +50 mV in Fig. 2J and K).”

Rectification is defined as the ratio of the currents measured at the positive and negative voltages of the same absolute magnitude. Indeed, Figures 2J and K show larger outward currents from KnRh3 than from CrChR2, but inward currents from the former are also larger than those from the latter, so the two channelrhodopsins appear to have similar degrees of inward rectification. The same conclusion can be drawn from comparison of panels G and H in Figure 7.

Response: We thank the reviewer for their insightful suggestions. We reanalyzed the data. I-V plots of CrChR2 and KnChR, which were created after normalizing the current amplitude (at +50 mV as 1.0)(Fig. S3). When the absolute magnitude at +50 and -50 mV are compared, the photocurrent of KnChR at -50 mV is -1.085, indicating almost linear relation between +50 and -50 mV. On the other hand, the photocurrent of CrChR2 at -50 mV is -2.6, showing inward rectification. This indicates two channelrhodopsins exhibit different rectification. For better visualization, we add this analysis in the Figure S3. We also modified the text accordingly (line 249-254).

Lines 357-358: “showed no significant difference in ΔE_{rev} among H⁺, Na⁺ and K⁺ (45~48 mV), indicating that permeability of these cations was almost the same (Fig. 7K).”

It is not clear what the Authors mean by “permeability” here. The permeability RATIO (relative permeability) can be derived using the Goldman-Hodgkin-Katz equation. The relative permeability depends not only on the ΔE_{rev} measured after substituting the test ion for the control ion, but also on the actual concentrations (activities) of the two ions (see e.g. [PMID: 1431803]). Taking into account that the Authors used ~1400000-fold higher concentrations of Na⁺ or K⁺ than of H⁺ in their bath solutions, KnRh3 actually exhibited far greater relative permeability for H⁺ than for the metal cations, as is typical of chlorophyte CCRs (see e.g. [PMID: 14615590]).

Response: We thank the reviewer for the valuable suggestions. And the reviewer is correct that we meant the permeability ratio, not permeability. Taken account the suggestion, we modified the text and added a table for comparing the permeability ratio between KnChR and CrChr2. We also added a relevant reference and modified the text in the manuscript (Lines 338-347).

Minor issues:

Line 23: “novel channelrhodopsin KnRh3”

The names of all so far identified channelrhodopsins have been abbreviated as ChRs or, more recently, as CCRs or ACRs for cation and anion channelrhodopsins, respectively. On the other hand, the abbreviation “Rh” is usually reserved for animal visual rhodopsins that form a different superfamily, so the Authors’ using it here will unavoidably confuse the readers. As the Authors have

clearly demonstrated cation channel activity of this protein, I suggest they use the abbreviations “KnCCR” or “KnChR” instead of KnRh3. I understand that the Authors would like to keep the protein name consistent with their earlier preprint (<https://www.preprints.org/manuscript/202009.0015/v1>), but they could simply explain here that KnRh3 is a synonym.

Response: We have accepted valuable suggestions. We rename it as KnChR in the revised manuscript.

Lines 49-58: “Microbial-type rhodopsins... and other ion-pumping rhodopsins.”

A general description of the entire superfamily of microbial rhodopsins is not relevant for this study, as KnRh3 is highly homologous to the known CCRs from green algae (Lines 186-187). Please delete it. Instead, background information on the cytoplasmic domains in channelrhodopsins would be useful to introduce the readers to the problem. To this end, please move the lines 366-378 from Discussion to Introduction.

Response: We have deleted a general information of microbial rhodopsins and instead moved background information on cytoplasmic domain (the lines 366-378 in the previous text) into the introduction section (lines 50-83 in the revised text).

Line 196: “His instead of Asp at position 96 was regarded as a characteristic of channelrhodopsins”

Please delete this statement as grossly outdated. This conclusion has been drawn from comparison of the four closely related ChRs from *Chlamydomonas* and *Volvox* that were identified first; since then, hundreds of new ChRs were discovered in which this residue is not conserved.

Response: Thank you for the suggestion. We have deleted the mentioned statements.

Lines 197-198: “this position was occupied by Ala, by D156 in GtACR1, while C128 in CrChR2 formed a hydrogen bridge (D-C pair or D-C gate) which altered the channel open lifetime”

In GtACR1, the homolog of C128 is C102, not D156, so it is unclear what the Authors wanted to say in this sentence. Moreover, it is unclear why they mention GtACR1 in the first place. Please clarify.

Response: We are sorry for confusing description. The sentences were re-structured and modified (lines 180-181).

Lines 230-231: “As shown in Fig 2 B-F, large photocurrents were observed in all the five KnRh3 variants”

Please delete the word “large”, because this statement contradicts that in Line 253: “the photocurrent amplitudes also differed among the nine variants (Fig. 2I).” The latter figure shows that indeed the currents from the variants comprising >300 residues were significantly smaller than those from the shorter constructs.

Response: Thank you for the suggestions. We have deleted relevant text in revised manuscript (lines 213-214).

Line 278: “compare the channel-off kinetics and amplitude (Fig. 3)”

I suggest to move Figure 3 to supplement because it actually shows control data (verification that the observed differences in the current kinetics are independent on the presence of the fluorescent tag).

Response: Fig. 3 is moved to Fig. S6 in the revised version.

Line 283: “In fact, eYFP constructs contain the ER-exporting signal peptide”

This should be explained from the beginning, i.e. in Line 211, to avoid confusing the readers.

Response: Edited as suggested. Please see Lines 192-195 in the revised manuscript.

Line 303: “The double mutant R287A/R291A reached about 100 ms”

It was not the mutant itself, but the t-off of its current decay that reached this value; please correct.

Response: We are sorry for the confusion. The same is corrected in the revised manuscript (lines 286-289).

Line 315: “spectrum with two λ_{\max} at 430 and 460 nm”

By definition, the maximum is the largest value, so there can only be one maximum of the spectrum. Figure 5 shows that the spectrum of KnRh3-272 photocurrents exhibits the maximum at ~460 nm and a shoulder at ~440 nm, whereas that of KnRh3-397 has the maximum at ~440 nm and a shoulder at ~460 nm. So, it appears that the truncation changes relative contributions of the two spectral forms, which seems to be worth discussing in the text.

Response: Thanks for the valuable comments. As suggested, we added relevant descriptions in the revised manuscript. Please see lines 297~301

Line 331: “KnRh3 permeates monovalent and divalent cations”

Cations permeate through the channel, not the other way around; please correct.

Response: Thank you for pointing out our grammatical error. Corrected in the relevant section of the revised manuscript (line 316)

Line 366: “Several ChRs also have a long C-terminus tail”

“Several” is an understatement here. To the best of my knowledge, ALL so far known channelrhodopsins comprise a few hundreds of amino acid residues in their cytoplasmic domains.

Response: We agreed and corrected. Please see lines 70-73 in the revised text.

Lines 393-394: “Modulation of photocycle kinetics by the cytoplasmic domain has never been proven in ion-transporting rhodopsin thus far”

While this is correct, perhaps the Authors would like to cite the study in which it was demonstrated that the cytoplasmic domain controls the intramolecular charge movements in *Anabaena* sensory rhodopsin [PMID: 17012323].

Response: Thank you for suggesting a relevant paper. We cited and mentioned the effect of the cytoplasmic domain of *Anabaena* sensory rhodopsin. Please see lines 367~369.

Lines 418-419: “The cation channelrhodopsin from Cryptophyte algae (DTD channelrhodopsin) exhibited different ratio of cation permeation”

This statement makes an impression that the higher permeability for metal cations relative to proton distinguishes cryptophyte CCRs from their chlorophyte counterparts. This is however not true: for example, PsChR from Ref. 19 showed a similar Na⁺/H⁺ permeability ratio to that of cryptophyte BCCRs.

Response: Thank you for pointing out an important issue. The reviewer is correct that PsChR showed a similar Na⁺/H⁺ ratio to that of BCCR. In both cases, CCR generate the highest permeability for H⁺ over metal cations as shown in the table 1 in the revised version. To avoid confusion, the text was corrected in the revised manuscript accordingly (Lines 340-343 and 387-391).

Table 1: For the bath solution named “NMG6.85” no NMG is listed among the ions. Do I assume correctly that it was 140 mM NMG, and the pH was adjusted with MES? If so, please correct.

Response: Thank you for pointing out our mistake. Corrected and edited revised version of the manuscript accordingly (Table S1).

Reviewer #2 (Remarks to the Author):

This manuscript identifies and characterizes a new cation channelrhodopsin (KnRh3) from a terrestrial alga (*Klebsormidium nitens*). The channelrhodopsins have found important application in the burgeoning field of optogenetics, though this particular example may not prove to be useful in this application due to its short wave (430-460nm) absorption, one of the bluest shifted channelrhodopsins characterized thus far. This is also one of, if not the first channelrhodopsin from a terrestrial alga to be characterized. The authors characterized the channel using patch clamp to measure ion current and kinetics of the channel. The authors show KnRh3 to be a relatively nonselective cation channel, showing channel activity for a variety of cations including H⁺, Na⁺, K⁺ and even divalent cations such as Ca²⁺. The most interesting characteristics of this new channel include its blue shifted spectrum and the fact that the c-terminal domain regulates the channel, which has not been previously observed in a channelrhodopsin ion channel. The kinetics of channel closure is modulated from 10 ms to over 100ms by the c-terminal domain. This is shown first by deletion analysis, with the effect then localized to two basic residues in the domain. They also found that truncation of the c-terminal domain also leads to substantially increased light sensitivity (line 318-329). It would be interesting to know if the light sensitivity is also affected by the two above mentioned residues. This point should be addressed in the manuscript.

Overall, the data looks to be well-measured and analyzed and supports the conclusions of the paper. The findings are novel and of interest to both the rhodopsin field and to a broader field of researchers interested in algal adaptation to terrestrial existence, a key step in the evolution of land plants.

Response: We thank the reviewer for the supportive comments regarding work and action/absorption spectra of KnChR (KnRh3).

Line 32: ...two Arginine residues, R287 and R291, that are crucial...

Response: Corrected and incorporated revised version of the manuscript accordingly. Please see lines 29-30 in the revised text.

Line 90: FimV is involved in peptidoglycan binding protein... (this sentence needs to be clarified)

Response: We have re-written it and incorporated revised version of the manuscript accordingly. Please see lines 69-70, and 404-416 in the revised text.

Line 95: light sensitivity and ion selectivity were compared...Line 148: ...fluorescent images were acquired with a fluorescent microscope...

Response: Corrected and incorporated revised version of the manuscript accordingly. Please see lines 137-138 in the revised text.

Line 241...this indicates that the channel kinetics and the photocycle are affected by the cytoplasmic domain...The authors should elaborate more to explain exactly how the observations demonstrate a change in the photocycle.

Response: Thank you for pointing out a critical issue. In ChR2 studies, it was demonstrated that the channel kinetics (channel closure) reflects the decay of the M-intermediate during the photocycle (Ritter et al. J. Biol. Chem. 283, 35033-41, 2008). Thus, we anticipated that this is also true in KnChR. (KnRh3). However, this is not Experimentally shown in our study, as the reviewer #1 and #3 mentioned. To measure the photocycle of KnChR by spectroscopic experiment, we have actually tried to express and isolate the protein. But no success of sufficient expression so far. Thus, it is currently difficult to assess the photocycle. Taken the reviewers suggestion #1 and #3, we delete the conclusion about the relation between the phtocycle and the cytoplasmic domain.

Line 361: ...in addition, ts was analyzed.... (the meaning here is not clear and should be rewritten)

Response: Thank you for the suggestion. We have re-written it and incorporated in revised version of the manuscript accordingly (lines 348-350).

Line 379: should read: Phosphorylation site prediction of KnRh3 suggested that several motifs may be involved in phosphorylation

Response: We have re-written it and incorporated in revised version of the manuscript accordingly (Lines 398-399).

Reviewer #3 (Remarks to the Author):

By genome screening the authors have identified a channelrhodopsin (ChR) called KnRh3 from an alga different in habitat (terrestrial) and morphology (filamentous) from those previously reported. There are three different properties of KnRh3 that the authors identify in their manuscript: (1) A unique structural feature is that following the membrane-inserted opsin domain, the ChR contains an FimV domain in the cytoplasmic C-terminal region. FimV is a peptidoglycan-binding protein found in gram-negative bacteria and has been shown to be involved in pilus assembly, migration of components of cells involved in cell-division, and motility, all of which are likely to involve FimV binding to the peptidoglycan layer found in gram-negative bacteria. Since algae do not have a peptidoglycan layer in their cell wall, the role of FimV and its association with a light-activated cation channel is mysterious and interesting.

(2) KnRh3 is only the second of the far-blue-shifted ChRs known, the first, PsChR, reported in a marine alga in 2013. This property is of interest to users of ChRs as optogenetic tools.

(3) The authors demonstrate that the cytoplasmic region interacts with the 7-TM opsin domain and slows channel closing, and present mutant data indicating that electrostatic interaction between the opsin domain and the cytoplasmic domain accelerates channel closing. No such effect was observed in measurements with opsin domain vs full-length CrChR2.

I believe that there is sufficient potential importance of KnRh3 to eventually publish an article on its properties listed above. The work presented in this submission provides a good start, but is in several respects preliminary. Before publication, the authors need to bring the level of understanding of the molecule, especially of aspects (1) and (2), beyond a brief scanning of properties followed by speculative ideas. I list comments regarding some concerns and identify additional discussion and results that could provide further knowledge about KnRh3 to complete the manuscript.

Comments:

1. The authors state several proposals and conclusions about the photocycle, such as “electrostatic interaction between the 7-TM domain and the C-terminal domain accelerates the photocycle” and “Thus, the apparent current amplitude of the short variants was elevated as the M-state accumulated during illumination”. Since the authors do not report any photocycle measurements in the submission, any comments regarding intermediates and their lifetimes are speculative and unjustified. Flash spectroscopy of the pigment would relate the photocycle to the ion currents.

Response: We thank the reviewer for the valuable suggestions. We have actually tried to express and isolate the protein to measure the photocycle by flash photolysis. But no success for obtaining sufficient functional expression so far. Thus, it is currently difficult to assess the photocycle of KnChR (KnRh3). Taken the reviewers words (#1 and #3), we delete the conclusion about the relation between the C-terminal domain and the photocycle. We only addressed the C-terminal length and the channel kinetics in the revised text.

2. The authors claim that “In addition, KnRh3 would expand the optogenetics tool kit, especially for when short wavelength excitation is required.” The value of a blue-shifted ChR for use in optogenetics was put forth when the first blue-shifted one, PsChR, was reported in 2013 (authors’ reference 19). This idea was confirmed and extended recently by Nagel and coworkers (Duan X, Nagel G, and Gao S. 2019. Mutated Channelrhodopsins with Increased Sodium and Calcium Permeability. *Appl. Sci.* 2019, 9(4), 664; <https://doi.org/10.3390/app9040664>). The authors need to describe prior published studies in terms of relevance to optogenetics and compare KnRh3 properties with those of PsChR. If the authors want

to claim that the blue-shifted KnRh3 would be useful as an optogenetics tool, they need to compare the efficacy of neuron activation with that previously published for the similarly blue-shifted PsChR. Are there advantages to KnRh3 over PsChR for optogenetics use? The authors compare properties of KnRh3 important for optogenetic tools, such as ion selectivity, with CrChR2, but the most relevant comparison would be to PsChR.

Response: We thank the reviewer for suggesting important papers. We considered and relevant facts have been incorporated in the revised manuscript. We have not tested KnRh3 (KnChR in the revised text) in neuronal cells. Thus, it is unfortunately not possible to compare the efficacy of neuron activation with PsChR. However, we described properties of CrChR2, PsChR and KnChR in terms of ion selectivity (permeability ratio). In particular, we added Table 1 for comparing the ion permeability ratio between CrChR2, PsChR, and KnChR. One of the advantages of KnChR is a relatively good Ca²⁺ permeability. KnChR-wt already shows high permeability ratio (Ca²⁺/Na⁺) as Catch (ChR2 L132C). Thus, it has a potential for engineering even higher Ca²⁺ permeable KnChR. We have described above in the results and the discussion part in the revised version. These changes would support applicability of KnChR for optogenetics. Please see lines 338-347, and 385-397.

3. The action spectrum may be influenced by other pigments with unknown absorption spectra altering the intensity of the actinic light. The shape of the action spectrum with 2 equal extinction peaks needs explanation. Is it due to vibrational fine structure? Two conformations of the pigment? Influence of unrelated pigments in the cells? The authors need to examine the absorption spectrum of the purified KnRh3. In vitro analysis of KnRh3 would also facilitate the flash spectroscopy (Comment 1) and comparison with PsChR needed (Comment 2).

Response: We thank the reviewer for the valuable comment. We think that absorption spectrum of KnChR consists of two isoforms with different absorption maxima. Such properties have been reported for several ChRs including CrChR1 and CrChR2 (Berthold et al. Plant Cell, 20, 1665–77, 2008, Ritter et al. J. Biol. Chem. 283, 35033-41, 2008). Therefore, such properties can be anticipated in KnChR. It is currently difficult to demonstrate it without purified protein. Furthermore, the flash spectroscopy experiment for comparison with PsChR cannot be performed. We are working for improving functional expression of KnChR. Thus, this point will be addressed in the future study. In the revised paper, we added some more description concerning the action spectrum which was also pointed out by reviewer #1 (Lines 396-301).

4. An unusual feature of KnRh3 is that it includes a FimV domain, a peptidoglycan found typically in gram-negative bacteria, at the C-terminus of rhodopsin. FimV is a peptidoglycan-binding protein which has been found to be involved in secretion, cell-division, and motility in gram-negative bacteria. The authors conclude that “it is anticipated” that light signals are transduced via ChR into motility in *K. nitens*. This conclusion is not justified. The processes (motility being only one of them) in gram-negative bacteria are likely due to FimV binding to the peptidoglycan layer. Since algae do not have a peptidoglycan layer in their cell wall, the role of FimV is an open, and very interesting, question. Peptidoglycan is found in some algae in their prokaryotic-derived photosynthetic organelles in their cytoplasm. Is peptidoglycan found in *K. nitens*? If so, is it located in the same compartment of the cell as the cytoplasmic domain of KnRh3? Does peptidoglycan bind to KnRh3 protein, and, if so, does it alter its channel properties? To start, the authors need to summarize the literature regarding peptidoglycan in algae.

Response: We are thankful to the reviewer for posing valuable questions that will help to refine our current manuscript. We agree with the reviewer that in the present scenario it is difficult to hypothesize the role of FimV only in motility and therefore its role in algae is an open question. We have modified the relevant sentence in our manuscript. Please see Lines 404-416 in the revised text.

When searched for literature regarding the presence of peptidoglycan in algae. We found papers suggesting presence of peptidoglycan layer around the chloroplast in glaucophyte and streptophyte (specifically in *Klebsormidium nitens*) of algal groups but not in chlorophyte group^{1,2}. Algal peptidoglycan layer is similar to peptidoglycan present in bacterial cell wall, since it was sensitive to antibiotic (penicillin) and D-cycloserin (inhibitor of D-Ala:D-Ala ligase) and affects the cell division of chloroplast². Presence of peptidoglycan biosynthetic pathway genes such as FtsZ3 (Filamentous temperature-sensitive Z) involved in chloroplast maintenance and chloroplast division was considered as a marker for presence of peptidoglycan layer and Mur genes³⁴. FtsZ3 showed its presence in glaucophyte and streptophyte but not in chlorophyte³. Along with FtsZ3, presence of Mur genes and sensitivity to antibiotic (effect on chloroplast division) was studied in algae and land plants. FtsZ3 and Mur genes were present in klebsormidiales and were also sensitive to antibiotic confirming the presence of peptidoglycan in *Klebsormidium nitens*³.

Our result indicates the presence of FimV domain of KnChR on the cytoplasmic side, and peptidoglycan layer is present around the chloroplast of *Klebsormidium nitens*.

Interaction of FimV of KnChR with peptidoglycan and its effect on channel activity needs further experimental evidence.

Relevant papers

1. Björn, L. O. Peptidoglycan in eukaryotes: Unanswered questions. *Phytochemistry* vol. 175 2019–2021 (2020).
2. Takano, H., Tsunefuka, T., Takio, S., Ishikawa, H. & Takechi, K. Visualization of Plastid Peptidoglycan in the Charophyte Alga *Klebsormidium nitens* Using a Metabolic Labeling Method. *Cytologia (Tokyo)*. 83, 375–380 (2018).
3. Grosche, C. & Rensing, S. A. Three rings for the evolution of plastid shape: a tale of land plant FtsZ. *Protoplasma* 254, 1879–1885 (2017).
4. Barreteau, H. *et al.* Cytoplasmic steps of peptidoglycan biosynthesis. *FEMS Microbiol. Rev.* 32, 168–207 (2008).

5. A general problem is that the manuscript is sometimes unclear because the authors use vague language. This problem starts from the beginning in their title. First, “unique light-gated ion channel properties”: What are the unique properties? There do not appear to be unique channel properties shown in the article. The aspect of the molecule that may be unique is the presence of a peptidoglycan binding domain (FimV), but no evidence is presented that FimV plays a role in activity of the channel. Second, “novel modular”: Essentially all channelrhodopsins are modular with cytoplasmic domains outside of the opsin domain, so the modularity itself is not novel. Third, the authors emphasize that the algal species encoding KnRh3 is an “evolutionary important alga”? The authors do not discuss any evolutionary implications of their findings, as would be expected from this claim being high-lighted in the title. The authors need to read through their manuscript and make sure that each sentence has a well-defined meaning. Such a

rewriting would greatly improve the manuscript and help readers understand the authors' interpretation of their data.

Response: We thank the reviewer for pointing out the unclear presentation of the data and facts. Taken the reviewer's words, we have changed the title. This was also suggested by the other reviewer. We have gone through the entire manuscript and re-phrased sentences for clarity to the best of our capacity.

REVIEWERS' COMMENTS:

Reviewer #1 (Remarks to the Author):

The Authors have much improved their manuscript during revision and taken into account all my concerns. I recommend the revised version for publication after a few minor changes are made, as listed below.

Line 32: "maximum action spectrum exhibited was at 430 nm and 460 nm"

Please change to "maximal sensitivity was exhibited at 430 and 460 nm"; the spectrum itself cannot "exhibit" anything.

Line 53: "generate permeability for"
Please change to "conduct".

Line 59: "ChR (ACR) was created"
Please change to plural: "ChRs (ACRs) have been created". There are ~100 natural ACRs already known.

Line 82: Please change "is connected to" to "regulates".

Line 88: Please delete "great": the utility of KnChR for optogenetics is yet to be tested.

Line 169: Please delete "Extensive and targeted" as self-evident.

Lines 181-183: "This pair is conserved in KnChR whereas chlorophyte cation channelrhodopsins (GtCCR1-4) lacks amino acid corresponding to D156".

Please delete or modify this sentence, because it is highly misleading: GtCCR1-4 not only differ from chlorophyte ChRs such as KnChR at this single position, but belong to an altogether different ChR family that shows very little overall homology to chlorophyte ChRs. On the other hand, there are many chlorophyte ChRs in which D156 is not conserved, such as CnChR1 (Chrimson) from *Chlamydomonas noctigama* or CbChR1 from *C. bilatus*.

Line 223: "with different lengths of amino acids"
Please change to "comprising different numbers of amino acid residues".

Lines 290-291: "These results clearly indicate that the positively charged residue, R287 and R291, interact with the 7-TM domain and contribute to altered channel kinetics"

The Authors provide no data demonstrating physical interaction between these residues and the 7TM domain. Therefore, "clearly indicate" needs to be changed to "strongly suggest".

Line 292: "these three mutants"
Please clarify, as only two residues are mentioned in Line 291.

Lines 316 and 336: Please change "generates" to "exhibits".

Line 320: "extracellular solution was systematically replaced with various cations"

Please add "sodium" as in "sodium in the extracellular solution was systematically replaced with various cations".

Reviewer #2 (Remarks to the Author):

The authors have satisfactorily answered the suggestions and critiques of the reviewers. The

manuscript is acceptable for publication.

Reviewer #3 (Remarks to the Author):

The manuscript is improved and those of my concerns that did not entail new measurements were satisfactorily addressed. However, my major comment was and still is that the work presented is interesting but too preliminary for Nature's Communications Biology. The potential for high impact findings exists either in a deeper understanding of the allosteric phenomenon in the title and/or the role of the FimV motif and its relationship to the photosignaling properties of the protein. Either of these require additional experiments.

Response to reviewers' comments

Reviewer #1 (Remarks to the Author):

The Authors have much improved their manuscript during revision and taken into account all my concerns. I recommend the revised version for publication after a few minor changes are made, as listed below.

Line 32: "maximum action spectrum exhibited was at 430 nm and 460 nm"

Please change to "maximal sensitivity was exhibited at 430 and 460 nm"; the spectrum itself cannot "exhibit" anything.

Answer: Changed. Please see line 36-37.

"Additionally, maximal sensitivity was exhibited at 430 nm and 460 nm, ..."

Line 53: "generate permeability for"

Please change to "conduct".

Answer: Changed. Please see line 47.

"These proteins conduct cations such as H⁺, Na⁺, K⁺, and Ca²⁺."

Line 59: "ChR (ACR) was created"

Please change to plural: "ChRs (ACRs) have been created". There are ~100 natural ACRs already known.

Answer: Changed. Please see line 53.

"Anion-conducting ChRs (ACRs) have been created artificially or naturally discovered."

Line 82: Please change "is connected to" to "regulates".

Answer: Changed. Please see line 76-77.

"Recently, it has been shown that phosphorylation of CrChR1 regulates photomotility and calcium signaling of green alga"

Line 88: Please delete "great": the utility of KnChR for optogenetics is yet to be tested.

Answer: Deleted. Please see line 82.

"Furthermore, KnChR has potential for expanding the optogenetics tool kit,..."

Line 169: Please delete "Extensive and targeted" as self-evident.

Answer: Deleted. Please see line 88.

“Mining of the genomic database of *K. nitens* revealed existence of several rhodopsin-encoding genes.”

Lines 181-183: “This pair is conserved in KnChR whereas chlorophyte cation channelrhodopsins (GtCCR1-4) lacks amino acid corresponding to D156”.

Please delete or modify this sentence, because it is highly misleading: GtCCR1-4 not only differ from chlorophyte ChRs such as KnChR at this single position, but belong to an altogether different ChR family that shows very little overall homology to chlorophyte ChRs. On the other hand, there are many chlorophyte ChRs in which D156 is not conserved, such as CnChR1 (Chrimson) from *Chlamydomonas noctigama* or CbChR1 from *C. bilatus*.

Answer: Modified. Please see lines 100-101.

“This pair is conserved in KnChR.”

Line 223: “with different lengths of amino acids”

Please change to “comprising different numbers of amino acid residues”.

Answer: Changed. Please see lines 143-144.

“., we created in addition four variants comprising different numbers of amino acid residues..”

Lines 290-291: “These results clearly indicate that the positively charged residue, R287 and R291, interact with the 7-TM domain and contribute to altered channel kinetics”

The Authors provide no data demonstrating physical interaction between these residues and the 7TM domain. Therefore, “clearly indicate” needs to be changed to “strongly suggest”.

Answer: Changed. Please see lines 209-210.

“These results strongly suggest that the positively charged residue, R287 and R291,...”

Line 292: “these three mutants”

Please clarify, as only two residues are mentioned in Line 291.

Answer: Clarified. Please see lines 211-213. The two single mutants (R287 and R291) and the double mutant (R287A/R291A) exhibited significantly larger photocurrents than KnChR-397 (Fig. 3g) without any change in their reversal potential (Fig. 3h).

Lines 316 and 336: Please change “generates” to “exhibits”.

Answer: Changed. Please see line 236.

“KnChR exhibits permeability for H⁺, monovalent and divalent cations.”

Line 320: “extracellular solution was systematically replaced with various cations”

Please add “sodium” as in “sodium in the extracellular solution was systematically replaced with various cations”.

Answer: Added. Please see line 239-240.

“Sodium in the extracellular solution was systematically replaced with various cations,...”